# *S100a4*[+] alveolar macrophages accelerate the progression of precancerous atypical adenomatous hyperplasia by promoting the angiogenic function regulated by fatty acid metabolism

Hong Huang[1,2†], Ying Yang[2†], Qiuju Zhang[3], Yongfeng Yang[2,4], Zhenqi Xiong[3], Shengqiang Mao[2], Tingting Song[2], Yilong Wang[2], Zhiqiang Liu[2], Hong Bu[1,5*], Li Zhang[2,4,6,7*], Le Zhang[3,8*]

[1]Institute of Clinical Pathology, Key Laboratory of Transplantation Engineering and Immunology, Ministry of Health, West China Hospital, Sichuan University, Chengdu, China; [2]Institute of Respiratory Health, Frontiers Science Center for Disease-related Molecular Network, West China Hospital, Sichuan University, Chengdu, China; [3]College of Computer Science, Sichuan University, Chengdu, China; [4]State Key Laboratory of Respiratory Health and Multimorbidity, West China Hospital, Sichuan University, Chengdu, China; [5]Department of Pathology, West China Hospital, Sichuan University, Chengdu, China; [6]Department of Pulmonary and Critical Care Medicine, West China Hospital, Sichuan University, Chengdu, China; [7]Precision Medicine Key Laboratory of Sichuan Province, West China Hospital, Sichuan University, Chengdu, China; [8]Key Laboratory of Systems Health Science of Zhejiang Province, Hangzhou Institute for Advanced Study, University of Chinese Academy of Sciences, Hangzhou, China

*For correspondence:
hongbu@scu.edu.cn (HB);
zhangli2809@wchscu.cn (LZ);
zhangle06@scu.edu.cn (LZ)

†These authors contributed equally to this work.

Competing interest: The authors declare that no competing interests exist.

## eLife Assessment

This is an **important** study demonstrating the importance of S100A4+ alveolar macrophages in the earlier stages of tumour development and suggesting a role in angiogenesis. As such this **convincing** study is of interest to cancer biologists focused on early tumour development and those interested in the development of therapeutics that may specifically target early cancers.

**Abstract** Lung cancer is preceded by premalignant lesions, and what factors drive this transformation and the potential regulatory mode in the context of tumor initiation remain to be elucidated. In the course of precancerous lesions in mice, we found a phasic shift in metabolic patterns. Macrophages are a heterogeneous cell population with high plasticity in the tumor microenvironment. Single-cell interaction and metabolic analyses highlighted a cellular state, *S100a4*[+] alveolar macrophages, which exhibited distinct fatty acid metabolic activity, such as palmitic acid metabolism, at the atypical adenomatous hyperplasia stage, accompanied by an angiogenic-promoting function in a pre-neoplastic setting of mice. These findings were reproducible in human single-cell transcriptomes and had been confirmed by histopathological staining and in vitro cell coculture assays. Taken together, the results from this study demonstrated that the *S100a4*[+] alveolar macrophage subset contributes to tumorigenesis by altering its metabolic state, suggesting that metabolic interventions

targeting this cell state in the early stage of disease may delay neoplastic transformation of the lung epithelium.

## Introduction

The ontology of lung cancer is a multi-gene-involved, multi-stage, long-term, and extremely complex pathological process (*Greaves and Maley, 2012*). Emerging evidence indicates that multistep tumorigenesis develops through a series of progressive pathologic changes known as preneoplastic or precursor lesions, which present prior to the onset of cancer (*Pich et al., 2022*). With the development of imaging technology and the application of molecular biomarkers for lung cancer screening in recent years, the detection rate of early lung cancer, especially precancerous lesions, has increased significantly. Some of the abnormalities in precancerous lesions are reversible; early diagnosis and guiding precancerous patients away from carcinogens or early intervention and blocking may reverse their further development to minimize the risk of cancer (*Pennycuick et al., 2020*; *Teixeira et al., 2019*; *Yatabe et al., 2011*).

As the most common histologic subtype of lung cancer, lung adenocarcinoma (LUAD) predominantly arises from alveolar type 2 (AT2) cells (*Zacharias et al., 2018*). For the classification of LUAD and its precursors, atypical adenomatous hyperplasia (AAH) is considered to be the first step in a continuum of histomorphologic changes in malignant adenocarcinomas and is in continuity with preinvasive adenocarcinoma in situ (AIS) in terms of morphological alterations, then microinvasive lesions termed minimally invasive adenocarcinoma (MIA), and eventually invasive adenocarcinoma (IA) (*Sivakumar et al., 2017*; *Travis et al., 2011*). However, the scarcity and difficult accessibility of adequate samples limit the investigation of the cellular and molecular landscape of LUAD precursors. Alternatively, studies also found AAH-like lesions in mouse models, as well as adenomas and AIS, prior to the appearance of LUAD (*Marjanovic et al., 2020*; *Weichert and Warth, 2014*). These models accurately mimic human lung precancerous lesions at the molecular and histopathological levels, providing well-suited materials for our study. Gradually shifting the target of lung cancer treatment from middle and advanced patients with clinical symptoms to asymptomatic patients with early or precancerous lesions is precisely the new concept of oncotherapy in the future (*Reynolds et al., 2023*; *Umar et al., 2012*).

The tumor microenvironment (TME) homeostasis is determined by close crosstalk within and across all cellular components, including malignant, immune, endothelial, and stromal cells (*Vitale et al., 2019*). Unveiling potential communications between malignant cells and TME underlying oncogenesis is critical for the future development of mechanism-informed, subset-targeted tumor immunotherapy strategies (*Feng and Gao, 2021*; *Wen et al., 2021*). Tumor-associated macrophages (TAMs) are a cell population with plasticity and heterogeneity in the TME. As an important player in tumor pathobiology, on the one hand, macrophages may be educated by neoplastic cells to provide a favorable microenvironment, supporting the malignant transformation, disease progression, metastasis, and resistance to treatment; and on the other, they may play a role in anti-tumor immunity (*Mantovani et al., 2017*; *Ruffell et al., 2012*; *Sica and Mantovani, 2012*). Part of such heterogeneity may be attributed to the diversity of phenotypic characteristics, metabolic patterns, and functional profiles of TAMs in response to environmental perturbations (*Cassetta and Pollard, 2018*).

Tumorigenesis relies on reprogramming of cellular metabolism, and metabolic mining of immune cells in the TME has expanded our understanding of tumor-associated metabolic alterations at different stages of tumor development (*Bian et al., 2020*; *Li et al., 2018*). Metabolic changes of TAMs in TME have been reported to be accompanied by phenotypic and functional changes (*Vitale et al., 2019*; *Xia et al., 2020*). The metabolic landscape of TAMs in the context of tumor initiation and the interdependent relationship of development, metabolism, and functional plasticity in TAMs remain largely unraveled. Exploring the major metabolic circuits by which TAMs remodel the TME and digging metabolic clues that affect the functional polarization of TAMs will contribute to the proposal of immunometabolic strategies that utilize TAMs for tumor prevention and therapy.

In order to simulate the process of human lung adenocarcinogenesis, we established a spontaneous LUAD mouse model and used single-cell RNA sequencing (scRNA-seq) to parse the metabolic changes of malignant epithelial cells and macrophages during the precancerous period, explore the metabolic heterogeneity and pro-tumor mechanism of alveolar macrophages, and probe the key links of metabolic pattern shifts whereby the alveolar macrophage subset conditions epithelial cell

transformation. In-depth understanding of the molecular and cellular mechanisms underlying the acquisition of aberrant phenotypes at precancerous and further carcinogenesis stages may accelerate the identification of precancerous metabolic intervention targets. Therefore, the diagnosis and prevention of lung cancer can be advanced to the stage of precancerous lesions, and measures can be taken to prevent or reverse the further development of precancerous lesions.

## Results

### Histopathology profiling and scRNA-seq of precancerous lesions in mice

A/J mice have the highest incidence of spontaneous lung tumors among various mouse strains, and this probability significantly increased with age (*Landau et al., 1998*). To more comprehensively mirror the tumor initiation and progression process of human lung cancer, A/J mice were maintained for 12–16 mo for spontaneous lesions, which resulted in three recognizable precancerous lesions in the lung. A total of 19 tissues were histologically categorized by two pathologists into four subtypes, including eight normal tissues, three AAHs, three adenomas, and five AISs, which were confirmed by Ki67 staining, spanning the cascade from normal to precancerosis (*Figure 1B and C*). To characterize cell diversity along tumorigenesis, we conducted a time-ordered single-cell transcriptomic profiling starting with normal tissue and ending with AIS (*Figure 1A*). For each sample, we isolated single cells without prior selection for cell types and used the 10 x Chromium platform to generate RNA-seq data. After removing low-quality cells, a total of 119,698 cells that passed quality control were retained for subsequent analysis, which yielded a median of 910 detected genes per cell. The cell count and genes expressed in each sample were provided in *Figure 1—figure supplement 1A, B*.

To identify distinct cell populations based on gene expression patterns, we performed dimensionality reduction and unsupervised cell clustering using methods implemented in Seurat, followed by removing batch effects among multiple samples. As shown using Uniform Manifold Approximation and Projection (UMAP), profiles along the cascade were derived, and a total of 13 main cell clusters were finally identified, which we defined as the single-cell transcriptome atlas of mouse lung in premalignant lesions (*Figure 1D*). Based on the expression of canonical markers, we classified immune cells into T cells (*Cd3e*, *Cd3d*, *Cd3g*), B cells (*Cd79a*, *Ms4a1*, *Igkc*), neutrophils (*Retnlg*, *S100a8*, *S100a9*), macrophages (*Mrc1*, *Cd68*, *C1qa*), monocytes (*F13a1*, *Ms4a6c*), dendritic cells (*H2-Eb1*, *H2-Aa*, *H2-Ab1*), and mast cells (*Csf1*, *Il6*) and identified five clusters of non-immune cells, including epithelial cells (*Sftpc*, *Scgb1a1*, *Sftpb*), fibroblasts (*Col1a1*, *Dcn*, *Gsn*), endothelial cells (*Cdh5*, *Ramp2*, *Cldn5*), mesothelials (*Gpm6a*, *Upk3b*, *Msln*), and smooth muscle cells (*Acta2*, *Tagln*, *Myh11*). In addition, a cluster expressing proliferating markers (*Top2a*, *Mki67*) was identified, named cell cycle cells (*Figure 1E*). Each of these populations was captured from different pathological stages of different mice (*Figure 1F*). The proportion of cell types across the four stages showed irregular changes, while transcriptional homogeneity was reduced with precancerous progression, illustrating the importance of heterogeneity in tumorigenesis and also proving the reliability of the sampling in this study (*Figure 1G*, *Figure 1—figure supplement 1C, D*).

### Identification of initiation-associated epithelial marker and resolution of metabolic pattern in malignant epithelial cells

Tumorigenesis relies on reprogramming of cellular metabolism. To obtain comprehensive cell type-specific metabolic profiles, metabolism-related genes were extracted from the Kyoto Encyclopedia of Genes and Genomes (KEGG) database, and metabolic clustering was performed on scRNA-seq data. The metabolic subsets and cell types corresponded well, indicating metabolic heterogeneity among cell types. Through Gene Set Enrichment Analysis (GSEA) against metabolic pathways, 16 subsets manifested distinct metabolic activities, with neutrophils, epithelial cells, and macrophages showing relatively higher metabolic activities (*Figure 2—figure supplement 1A–C*).

Generally, malignant transformation of epithelial cells is the starting point for the development of precancerous lesions. AT2 cells, ciliated cells, and AT1 cells were identified in our epithelial transcriptome (*Figure 2A*). At the evolutionary stages of precancerous LUAD, despite possible sample heterogeneity and other interference, we observed increased interactions between epithelial cells and surrounding stromal and immune cells in the microenvironment, indicating gradually frequent

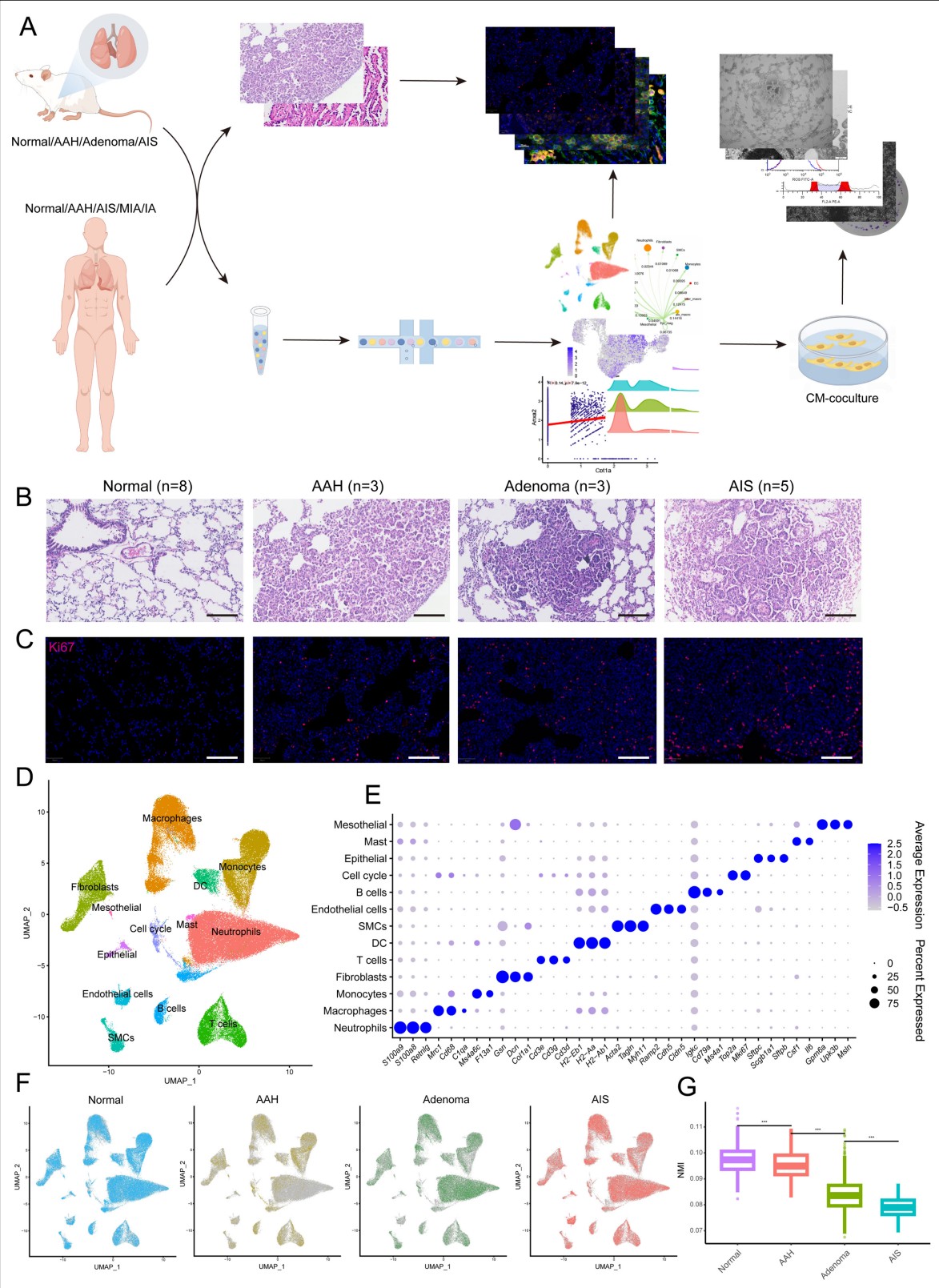

**Figure 1.** Histopathological grading and single-cell transcriptome profiles of mouse lung adenocarcinoma (LUAD) precancerous lesions. (**A**) Analysis and experimental flow chart of this study. (**B**) Hematoxylin and eosin (H&E) images of normal tissue and three stages of precancerous lesions (AAH, adenoma, and AIS) in mice. Scale bar: 100 μm. (**C**) Immunofluorescence staining of Ki67 at the four histopathological stages. Scale bar: 100 μm. (**D**) Uniform Manifold Approximation and Projection (UMAP) plot of 13 cell types from mouse single-cell RNA sequencing (scRNA-seq) data. (**E**) Dotplot

*Figure 1 continued on next page*

*Figure 1 continued*

of marker genes in all cell types. (**F**) Cell distribution at the four stages. (**G**) Reduced transcriptional homogeneity with progression of precancerous lesions. Transcriptional heterogeneity between cells is inversely proportional to normalized mutual information (NMI). AAH: atypical adenomatous hyperplasia; AIS: adenocarcinoma in situ; MIA: minimally invasive adenocarcinoma; IA: invasive adenocarcinoma; CM: conditioned medium; DC: dendritic cell; SMC: smooth muscle cell; ***p<0.001.

The online version of this article includes the following figure supplement(s) for figure 1:

**Figure supplement 1.** Quality assessment and cell fraction analysis of mouse single-cell RNA sequencing (scRNA-seq) data.

cell-cell communication during this process (*Figure 2B*). It was worth noting that the *Spp1-Cd44* complex showed more interactions in the AAH stage than in the AIS stage, between epithelial cells and other cell types, especially myeloid cells (macrophages, monocytes, and neutrophils). The SPP1-CD44 axis was reported to be involved in tumor initiation and growth, and CD44 levels were found to be higher in AAH compared to AIS and IA (*Kerr et al., 2004*; *Pietras et al., 2014*). In addition, the pattern of the *App-Cd74* signaling suggested incremental communications between epithelial cells and immune cells during the occurrence of precancerous lesions, and it was reported to be implicated in tumor progression and metastasis (*Anderson et al., 2023*). Then we sought to identify initiation-associated epithelial markers underlying LUAD precancerosis. *Ldha* was gradually enriched in the precancerous occurrence of lung cancer, and the increased expression trend along with the tumor initiation process was verified by immunofluorescent staining (*Figure 2C*, *Figure 2—figure supplement 1D*). Moreover, analysis of The Cancer Genome Atlas (TCGA) database indicated that the initiation-associated marker *LDHA* was highly expressed in LUAD and showed an increasing trend in the early stages of the disease, and its high expression was correlated with poor prognosis of LUAD patients (*Figure 2D–F*). Thus, it deserves to be further evaluated as a potentially promising biomarker during lung tumorigenesis.

To distinguish malignant cells from non-malignant cells based on karyotypes, chromosomal copy number variations (CNVs) from each epithelial cell's profile were inferred (*Figure 2G*, *Figure 2—figure supplement 1E*). There were some CNV variations in the normal group, which may be related to the nature of the algorithm, the sample collection operation, or the late processing of the cells; these cells were excluded from subsequent analyses. To explore the phasic metabolic changes in malignant epithelial cells, metabolic activity was quantified using the scMetabolism package. The results indicated that malignant epithelial cells in the adenoma stage were enriched in lipid metabolism-related pathways, while in the AIS stage, significant enrichment of glycometabolism-related pathways was observed (*Figure 2H*). The Gene Set Variation Analysis (GSVA) enrichment analysis showed a concordant variation trend (*Figure 2I*). These data suggested that epithelial cells manifest diverse metabolic patterns during malignant transition.

## $S100a4^+$ alv-macro exhibited active fatty acid metabolism in the AAH phase

With the development of tumor metabolism research, it has been found that the metabolism of other cell types in the microenvironment besides epithelial cells can also regulate tumorigenesis. Macrophage was the most prominent cell type that interacted with malignant epithelial cells, as shown by Cell Chat analysis. Macrophages were composed of alveolar macrophages (*Itgax* and *Siglecf*) and interstitial macrophages (*Itgam* and *Cx3cr1*) in subclustering (*Figure 3—figure supplement 1A–C*). Further analysis revealed a higher correlation between alveolar macrophages and malignant epithelial cells (*Figure 3A*, *Figure 3—figure supplement 1D*). To resolve the metabolic pattern of alveolar macrophages at a higher resolution, we subclassified them into four subpopulations based on gene expression (*Figure 3—figure supplement 1E, F*). Enrichment analysis was performed for each of the four subpopulations, and scMetabolism analysis revealed that one of the alveolar macrophage subpopulations already showed a metabolically active state at the AAH stage, especially lipid metabolism, which we hereafter termed as $S100a4^+$ alv-macro (*Figure 3B and C*). As a representative, the overall abundance trends of glycerolipid metabolism, glycerophospholipid metabolism, and biosynthesis of unsaturated fatty acids were displayed in *Figure 3D*. Next, we used the Compass algorithm to model the metabolic flux of cells for $S100a4^+$ alv-macro, quantitatively analyzed the metabolic activity of cells, and then focused on metabolic enzymes and genes related to lipid metabolism for

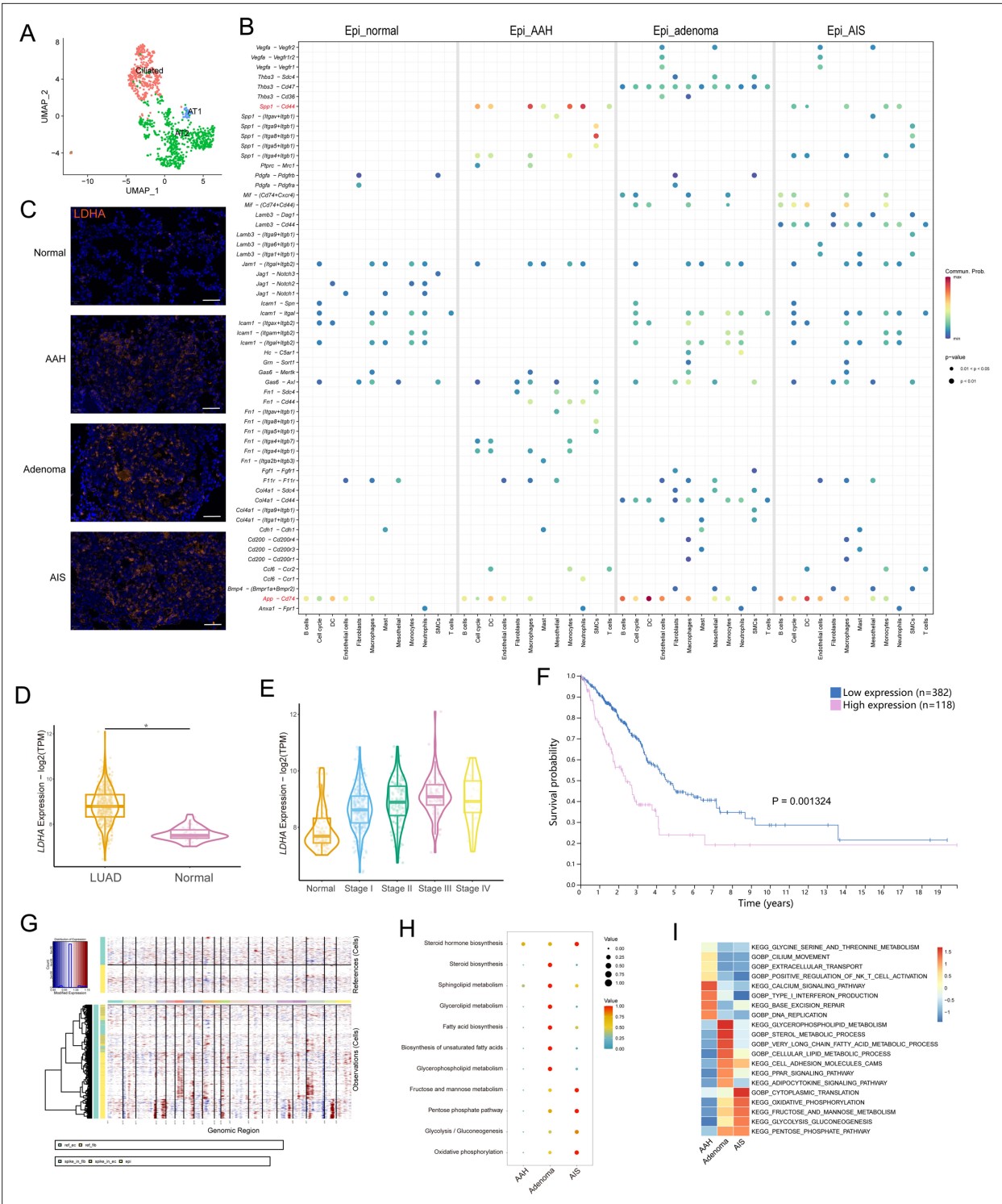

**Figure 2.** Identification of initiation-associated epithelial marker *Ldha* and analysis of malignant epithelial cells. (**A**) Uniform Manifold Approximation and Projection (UMAP) plot of epithelial cell subtypes. (**B**) Dotplot showing the significance (p-value) and strength (communication probability) of specific interactions between epithelial cells and other cell types at the four stages. Data was obtained by Cell Chat analysis. (**C**) Immunofluorescence staining of LDHA at the four stages. Scale bar: 50 μm. (**D**) *LDHA* expression in LUAD and normal tissues from the The Cancer Genome Atlas (TCGA) database. (**E**) *LDHA* expression in stage I-IV LUAD and normal tissues from the TCGA database. (**F**) Correlation of *LDHA* expression with overall survival of LUAD patients in the TCGA database. (**G**) Copy number variations (CNVs) (red, amplifications; blue, deletions) across the chromosomes (columns) inferred from the single-cell RNA sequencing (scRNA-seq) of each epithelial cell (rows). (**H**) scMetabolism analysis of malignant epithelial cells at precancerous

*Figure 2 continued on next page*

*Figure 2 continued*

stages. (**I**) Gene Set Variation Analysis (GSVA) enrichment analysis of malignant epithelial cells at precancerous stages. AT1/2: type I/II alveolar epithelial cells; LUAD: lung adenocarcinoma; *p<0.05.

The online version of this article includes the following figure supplement(s) for figure 2:

**Figure supplement 1.** Single-cell metabolic state clustering.

in-depth analysis. Fatty acid metabolism was found to be relatively more active in the AAH phase, which was represented by changes in palmitoyl-CoA desaturase, palmitoyl-CoA hydrolase, and carnitine palmitoyl-transferase (*Figure 3E*). Consistently, the expression levels of related genes *Cpt1a* and *Acot2* were also higher in precancerous lesions relative to normal circumstances (*Figure 3F*). In our preliminary validation, the presence of F4/80$^+$/CD11c$^+$/S100A4$^+$ alv-macro was identified in the corresponding tissues by multiplexed immunohistochemistry staining, and their number and proportion were observed to be significantly greater in AAH samples than in normal samples (*Figure 3G*, *Figure 3—figure supplement 1G, H*). CPT1A is responsible for the fatty acid β-oxidation in mitochondria (*Nakamura et al., 2014*), and its positivity was found more frequently in the *S100a4$^+$* alv-macro of the AAH group (*Figure 3G*), suggesting fatty acid metabolism in this subpopulation was activated at the AAH stage.

## Pro-angiogenic function of *S100a4$^+$* alv-macro was associated with fatty acid metabolism

Metabolic profile is coupled with phenotype and functional program of macrophages. Among all pathological stages, although there was no significant difference in the cell proportion of *S100a4$^+$* alv-macro in macrophages, its proportion in alveolar macrophages was obviously the highest in the AAH stage (*Figure 4A*, *Figure 4—figure supplement 1A*). To evaluate changes in the functional programs of this macrophage subpopulation in the context of precancerous lesions, we performed an enrichment analysis of the well-known macrophage phenotypes. It was revealed that the capacities for angiogenesis, M2-like polarization, and immunosuppression were found to be stronger in AAH or other precancerous stages relative to the normal stage (*Figure 4B*). The pro-angiogenic function predominated in the AAH stage, while M2-like and immunosuppressive functions were more prominent in the subsequent precancerous progression. Correlation analysis was used to measure the relationship between these functional programs and metabolic activity, where pro-angiogenic and M2-like phenotypes of *S100a4$^+$* alv-macro were proved to be positively correlated with lipid metabolism (*Figure 4—figure supplement 1B*). Further investigation showed that both of them were positively correlated with palmitic acid-related metabolic reactions (DESAT16_2_pos, C160CPT1_pos, and RE0577C_pos, *Figure 4C*). Next, we queried the angiogenesis-related gene signature in *S100a4$^+$* alv-macro, and the expression levels of *Anxa2* and *Ramp1* were detected to be higher in the AAH group than the normal group (*Figure 4D*). ANXA2 supports angiogenesis in specific tumor-related settings (*Huang et al., 2022*), and its expression level was positively correlated with that of *Cpt1a* in *S100a4$^+$* alv-macro (*Figure 4E*). What's more, the correlation between CPT1A and ANXA2 was verified in the immunostaining of AAH tissues, revealing more F4/80$^+$/CD11c$^+$/S100A4$^+$/CPT1A$^+$/ANXA2$^+$ alv-macro compared to normal tissues (*Figure 4F*). These results suggested that the pro-angiogenic function of *S100a4$^+$* alv-macro might be related to its fatty acid metabolic status.

## *S100a4$^+$* alv-macro displayed the potential to promote the early malignant transformation of lung epithelial cells in vitro

To further validate our findings from scRNA-seq and histological staining, *S100a4*-overexpressed (OE) alveolar macrophages were established by transfection of the *S100a4* vector into the murine MH-S cell line, and an empty vector was transfected as negative control (NC) cells, which were verified by quantitative real-time polymerase chain reaction (qPCR) and western blotting (*Figure 5A and B*). Then coculture assays were conducted to functionally prove the protumorigenic characteristics of *S100a4$^+$* alv-macro in vitro. After culture with conditioned medium (CM) of *S100a4*-OE MH-S, the proliferation and migration ability of mouse lung epithelial MLE12 cells were significantly promoted, and the cell cycle was more arrested in the G2/M phase (*Figure 5C–F*). Reactive oxygen species (ROS) produced within cells can lead to genetic mutations that trigger malignant transformation of cells

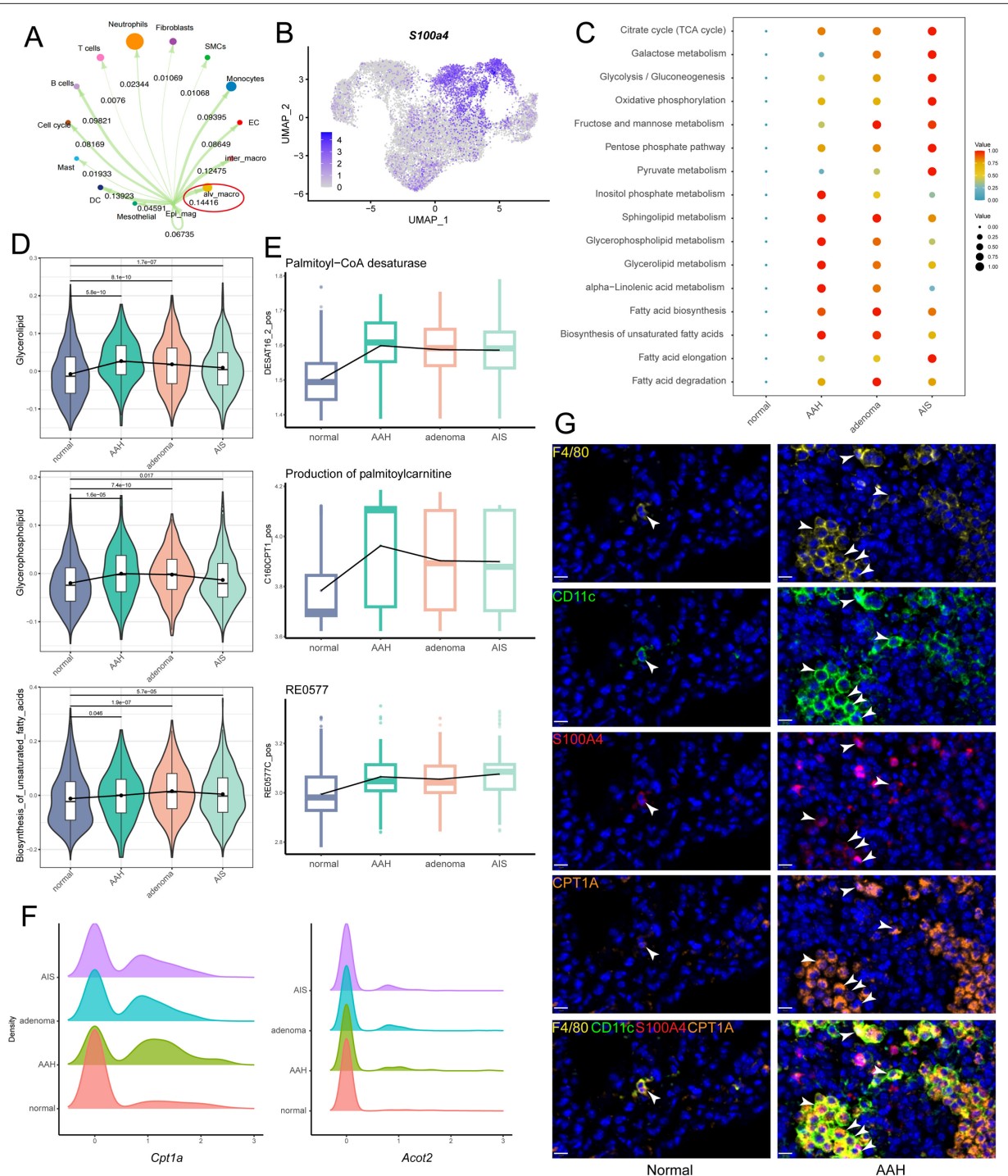

**Figure 3.** *S100a4*+ alv-macro was active in lipid metabolism at the adenomatous hyperplasia (AAH) stage. (**A**) Malignant epithelial cells showed the strongest communication weight with alveolar macrophages, as shown by Cell Chat analysis. (**B**) Uniform Manifold Approximation and Projection (UMAP) plot of *S100a4* expression in alveolar macrophages. (**C**) scMetabolism analysis of *S100a4*+ alv-macro at the four stages. (**D**) Changes in scores of representative lipid metabolism-related gene sets across the four stages. (**E**) Compass analysis showing the reaction activities of fatty acid metabolism across the four stages. (**F**) Density plots of *Cpt1a* and *Acot2* at the four stages. The x-axis represents the gene expression level, and the y-axis represents the density of numbers of cells. (**G**) Multiplex immunofluorescence validation of F4/80+/CD11c+/S100A4+ alv-macro in mouse normal and AAH tissues and comparison of tissue expression of CPT1A in this subpopulation. Scale bar: 10 μm. EC: endothelial cell.

The online version of this article includes the following figure supplement(s) for figure 3:

**Figure supplement 1.** Identification of *S100a4*+ alv-macro.

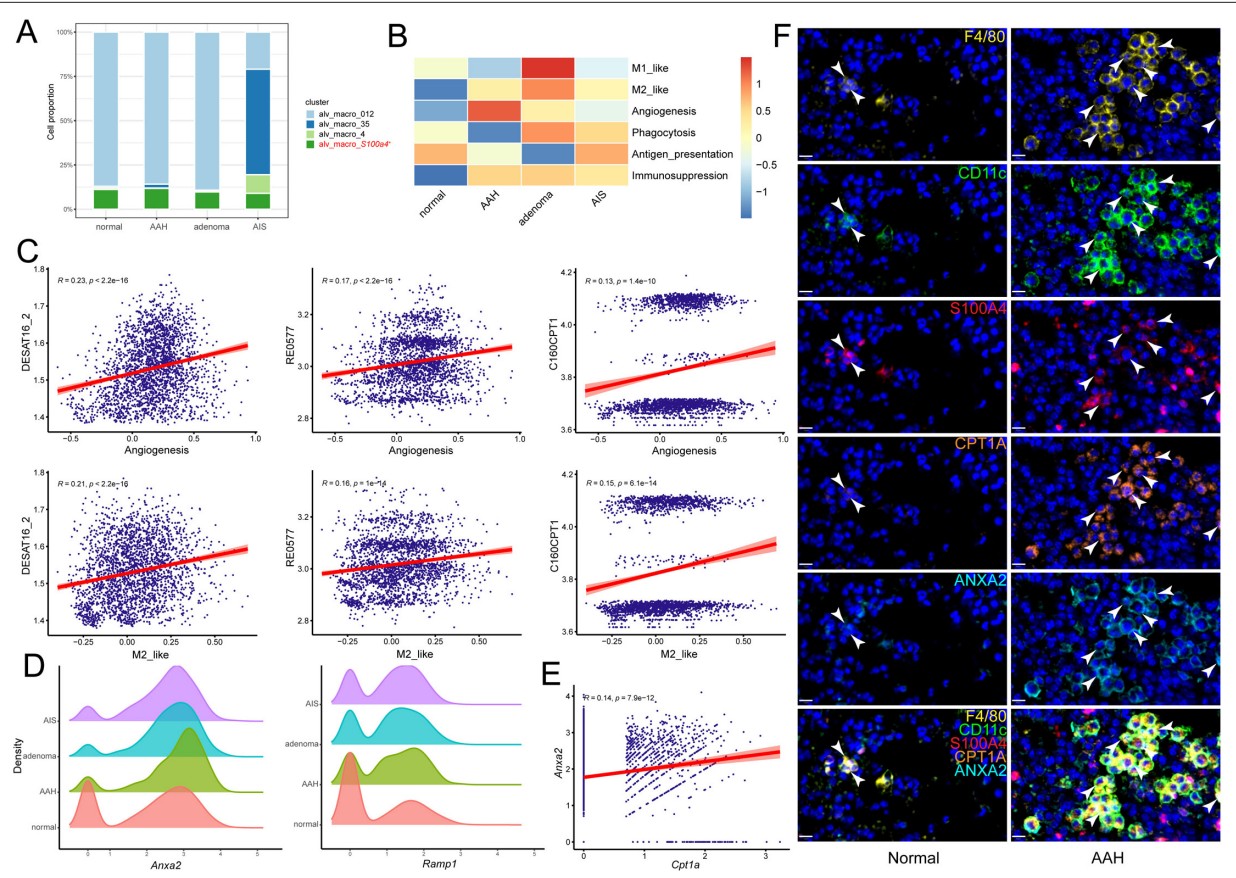

**Figure 4.** The pro-angiogenic function of *S100a4*+ alv-macro was related to fatty acid metabolism. (**A**) Cell proportion comparison of *S100a4*+ alv-macro in alveolar macrophages. (**B**) Macrophage functional program analysis of *S100a4*+ alv-macro across the four stages. (**C**) Pearson correlation analysis of angiogenesis and M2-like function with fatty acid metabolic reactions in *S100a4*+ alv-macro. (**D**) Density plots of *Anxa2* and *Ramp1* at the four stages. (**E**) Correlation analysis of *Cpt1a* and *Anxa2* expression in *S100a4*+ alv-macro. (**F**) Multiplex immunofluorescence validation of the correlation between CPT1A and ANXA2 expression in F4/80+/CD11c+/S100A4+ alv-macro of mouse normal and AAH tissues. Scale bar: 10 μm.

The online version of this article includes the following figure supplement(s) for figure 4:

**Figure supplement 1.** Correlation analysis between lipid metabolism and macrophage functions in *S100a4*+ alv-macro.

(*Santibáñez-Andrade et al., 2023*), and we found markedly elevated ROS levels in OE-CM cocultured epithelial cells (*Figure 5G*). Mitochondrial morphological change is one of the organelle indications of malignant transformation of cells (*Missiroli et al., 2020*). As shown in the electron microscope images, compared with the cocultured epithelial cells in the NC group, we observed the acquisition of irregular shapes, obvious vacuoles, and disorganized cristae in mitochondria of the OE group, which might be associated with energy metabolism in the process of cell transformation (*Figure 5H*). In addition, we performed reverse verification with siRNA and demonstrated that the functional phenotype of epithelial cells did not change after *S100a4*-knockdown MH-S coculture (*Figure 5—figure supplement 1*).

During the in vitro carcinogenesis process, a battery of assays allows for the identification of the cells in the initiation, promotion, or aggressive stages of tumorigenesis (*Barguilla et al., 2023*). We found increased expression of DNA damage marker p-γH2AX and decreased expression of adhesion protein E-Cadherin, but no marked changes in N-Cadherin, Vimentin, and stem-like markers (CD44 and CD133) (*Figure 5I*). The expression of tumorigenesis-associated proteins (c-MYC, p-ERK, SPA, VEGF, and HIF-1α) was elevated in OE-CM-cultured MLE12 (*Figure 5J*). Combined with the above observations and the absence of difference in invasion potential, we speculated that the coculture partially activated the epithelial-mesenchymal transition (EMT) program, and the transformation of lung epithelial cells was still in the early or intermediate stage and had not yet reached the advanced stage, which validated our findings to a certain extent and might correspond to the AAH stage in histology. In addition, the expression of pro-tumor indicators in macrophages (VEGF, MMP9, TGF-β,

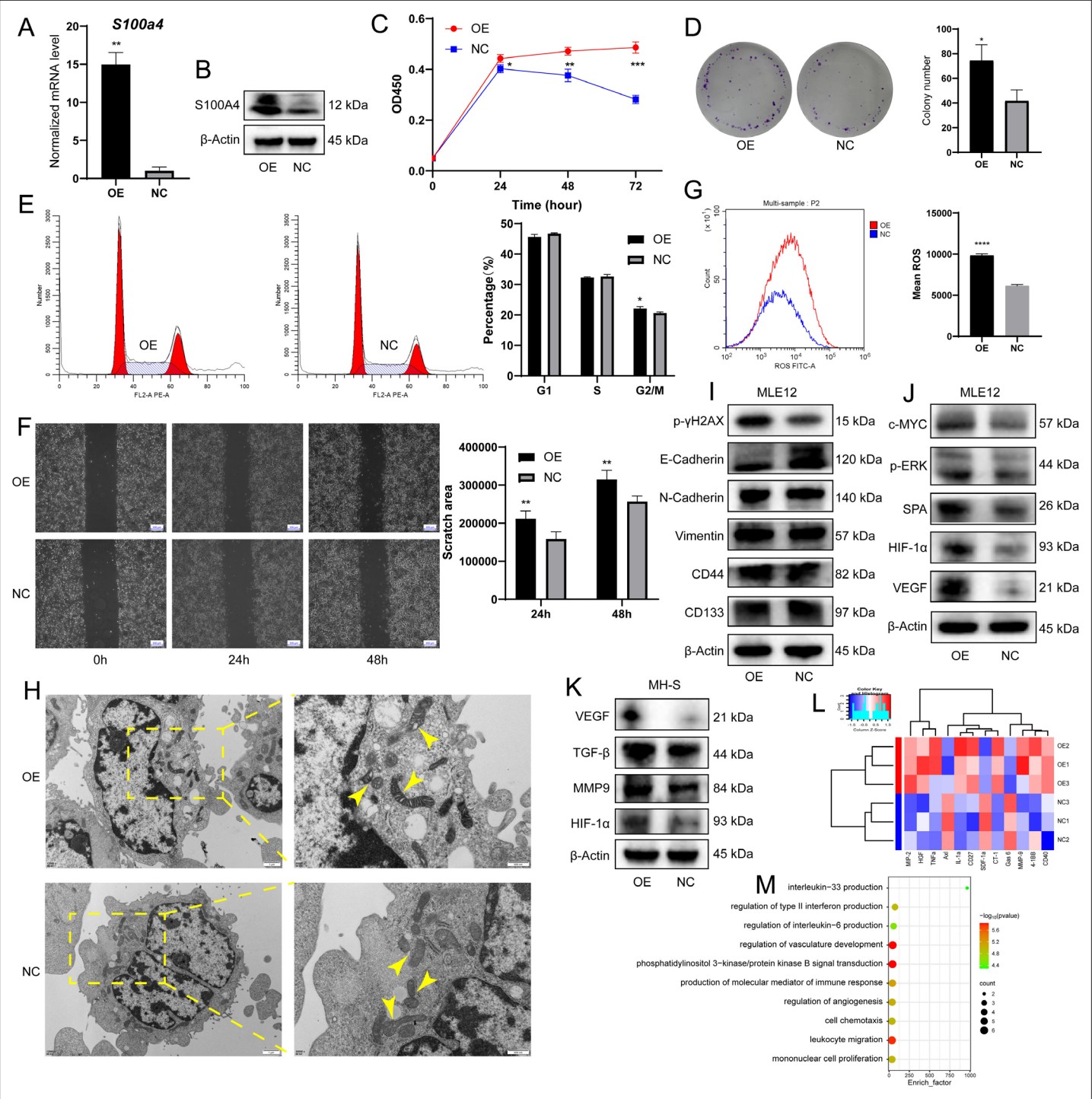

**Figure 5.** *S100a4*-OE MH-S promoted malignant transformation of MLE12 epithelial cells in vitro. (**A**) *S100a4* mRNA expression level in MH-S after transfection, n = 5. (**B**) S100A4 protein expression level in MH-S after transfection. (**C**) CCK8 assay showing cell proliferation of MLE12 after coculture with *S100a4*-OE MH-S, n = 6. (**D**) Colony-forming ability of MLE12 after coculture, as shown by the colony formation assay, n = 3. (**E**) Cell cycle distribution of MLE12 after coculture, as shown by cell cycle analysis, n = 3. (**F**) Wound healing assay showing cell migration of MLE12 after coculture, n = 3. Scale bar: 200 μm. (**G**) Intracellular ROS level of MLE12 after coculture, n = 3. (**H**) Transmission electron microscopy showing the morphological changes of mitochondria in MLE12 after coculture. Scale bar: 1 μm and 500 nm. (**I**) Western blotting of DNA damage marker p-γH2AX, EMT markers (E-Cadherin, N-Cadherin, and Vimentin), and stem-like markers (CD44 and CD133) in MLE12 after coculture. (**J**) Western blotting of tumorigenesis-associated proteins (c-MYC, p-ERK, SPA, VEGF, and HIF-1α) in MLE12 after coculture. (**K**) Western blotting of macrophage pro-tumor indicators (VEGF, MMP9, TGF-β, and HIF-1α) in *S100a4*-OE MH-S. (**L**) Heatmap of cytokine assay for differential factors secreted after MH-S transfection. (**M**) Enrichment

*Figure 5 continued on next page*

Figure 5 continued

analysis of GO biological processes for differential factors. The results shown above represent three or more replicates. OE: overexpression; NC: negative control; ROS: reactive oxygen species; *p<0.05, **p<0.01, ***p<0.001, ****p<0.0001.

The online version of this article includes the following source data and figure supplement(s) for figure 5:

Source data 1. PDF file containing original western blots for *Figure 5B*, indicating the relevant bands.

Source data 2. Original files for western blot analysis displayed in *Figure 5B*.

Source data 3. PDF file containing original western blots for *Figure 5I*, indicating the relevant bands.

Source data 4. Original files for western blot analysis displayed in *Figure 5I*.

Source data 5. PDF file containing original western blots for *Figure 5J*, indicating the relevant bands.

Source data 6. Original files for western blot analysis displayed in *Figure 5J*.

Source data 7. PDF file containing original western blots for *Figure 5K*, indicating the relevant bands.

Source data 8. Original files for western blot analysis displayed in *Figure 5K*.

Figure supplement 1. *S100a4*-knockdown MH-S did not promote the malignant transformation of MLE12 epithelial cells in vitro.

Figure supplement 1—source data 1. PDF file containing original western blots for *Figure 5—figure supplement 1B*, indicating the relevant bands.

Figure supplement 1—source data 2. Original files for western blot analysis displayed in *Figure 5—figure supplement 1B*.

and HIF-1α) was also higher in *S100a4*-OE MH-S (*Figure 5K*). Next, we detected tumor-inducing factors secreted by these unique macrophages using the Cytokine Antibody Array. We noted the production of macrophage inflammatory protein (MIP)–2, hepatocyte growth factor (HGF), tumor necrosis factor α (TNF-α), IL-1α, MMP9, and CD40, and these cytokine-related biological processes were mainly involved in the regulation of angiogenesis and immune response (*Figure 5L and M*). In summary, we demonstrated in vitro that *S100a4*[+] alv-macro promotes the early malignant transformation of lung epithelial cells by secreting tumor-promoting cytokines.

## *S100a4*[+] alv-macro drove angiogenesis by promoting *Cpt1a*-mediated fatty acid metabolism

Next, we sought to identify the metabolic and functional changes found in the AAH phase mentioned above. It was demonstrated that the pro-tumor polarization of macrophages is dependent on the accumulation of intracellular lipid droplets as a stable source of fatty acid metabolism (*Wu et al., 2019*). As expected, in *S100a4*-OE MH-S, we observed an increase in lipid droplet content, as well as enhanced function of mitochondria where fatty acid β-oxidation takes place (*Figure 6A*). The pro-angiogenic effect of this subset was confirmed by tube formation assay of human umbilical vein endothelial cells (HUVECs) (*Figure 6B*). Moreover, fatty acid metabolism-related CPT1A and angiogenesis-related ANXA2 were also up-regulated in *S100a4*-OE MH-S (*Figure 6C and D*). ANXA2 is considered to be a cellular receptor for plasminogen (PLG), often expressed on the surface of endothelial cells, macrophages, and tumor cells, which activates a cascade of pro-angiogenic factors by promoting the conversion of PLG to plasmin, thereby promoting angiogenesis and tumor progression (*Semov et al., 2005*; *Sharma, 2019*). We found synergistic upregulation of ANXA2 and PLG expression in *S100a4*-OE MH-S and cocultured HUVEC and MLE12, which may help explain how ANXA2 induction was involved in angiogenesis and malignant transformation (*Figure 6D*).

Then we aimed to figure out the contribution of *Cpt1a* to angiogenesis and pro-tumor function in *S100a4*[+] alv-macro. Following the addition of the CPT1A inhibitor etomoxir (ETO) to *S100a4*-OE MH-S, the cocultured MLE12 showed restoration of cell proliferation, migration, and ROS production, and the cocultured HUVEC also exhibited decreased tube-forming capacity (*Figure 6E–H*). At the protein expression level, we also identified the recovery of angiogenesis-related proteins in MH-S and cocultured MLE12, as well as a regression in the levels of pro-tumorigenic and M2 markers in MH-S (*Figure 6I and J*). It was also found that the effect of CPT1A may be dependent on the activation of PPAR-γ signaling. Therefore, *S100a4*[+] alv-macro may facilitate angiogenesis through CPT1A-PPAR-γ-mediated fatty acid metabolism, thereby triggering tumorigenesis.

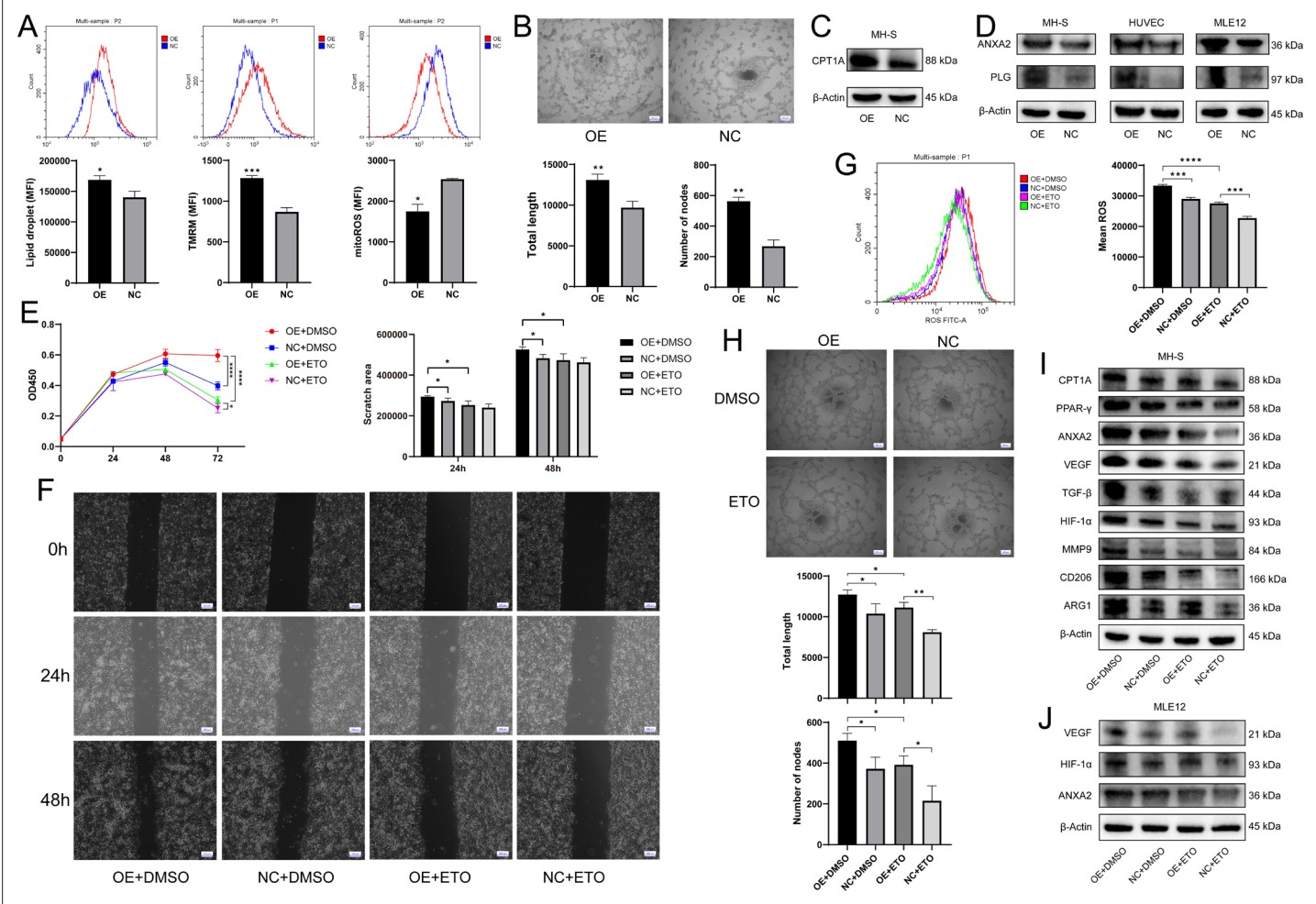

**Figure 6.** *S100a4-OE MH-S* controlled pro-angiogenic and pro-tumorigenic functions through *Cpt1a* induction. (**A**) Determination of intracellular lipid droplet accumulation, mitochondrial membrane potential, and mitochondrial reactive oxygen species (ROS) by flow cytometry, n = 3. (**B**) Tube formation of human umbilical vein endothelial cells (HUVECs) after coculture with *S100a4*-OE MH-S, quantified by total tube length and number of nodes, n = 5. Scale bar: 200 μm. (**C**) Western blotting of fatty acid metabolism-related CPT1A in *S100a4*-OE MH-S. (**D**) Western blotting of ANXA2 and PLG in *S100a4*-OE MH-S and cocultured HUVEC and MLE12. (**E**) CCK8 assay of cocultured MLE12 after treatment of ETO to *S100a4*-OE MH-S, n = 6. (**F**) Wound healing assay of cocultured MLE12 after treatment of ETO to *S100a4*-OE MH-S, n = 3. Scale bar: 200 μm. (**G**) Intracellular ROS production of cocultured MLE12 after treatment of ETO to *S100a4*-OE MH-S, n = 3. (**H**) Tube formation of cocultured HUVECs after treatment of ETO to *S100a4*-OE MH-S, n = 3. Scale bar: 200 μm. (**I**) Western blotting of fatty acid metabolism-related proteins (CPT1A and PPAR-γ), angiogenesis-related proteins (ANXA2, VEGF, TGF-β, HIF-1α, and MMP9), and M2 polarization markers (CD206 and ARG1) in *S100a4*-OE MH-S after treatment of ETO. (**J**) Western blotting of angiogenesis-related proteins (ANXA2, VEGF, and HIF-1α) in cocultured MLE12 after treatment of ETO to *S100a4*-OE MH-S. The results shown above represent three or more replicates. MFI: mean fluorescence intensity; ETO: etomoxir; *p<0.05, **p<0.01, ***p<0.001, ****p<0.0001.

The online version of this article includes the following source data for figure 6:

**Source data 1.** PDF file containing original western blots for *Figure 6C*, indicating the relevant bands.

**Source data 2.** Original files for western blot analysis displayed in *Figure 6C*.

**Source data 3.** PDF file containing original western blots for *Figure 6D*, indicating the relevant bands.

**Source data 4.** Original files for western blot analysis displayed in *Figure 6D*.

**Source data 5.** PDF file containing original western blots for *Figure 6I*, indicating the relevant bands.

**Source data 6.** Original files for western blot analysis displayed in *Figure 6I*.

**Source data 7.** PDF file containing original western blots for *Figure 6J*, indicating the relevant bands.

**Source data 8.** Original files for western blot analysis displayed in *Figure 6J*.

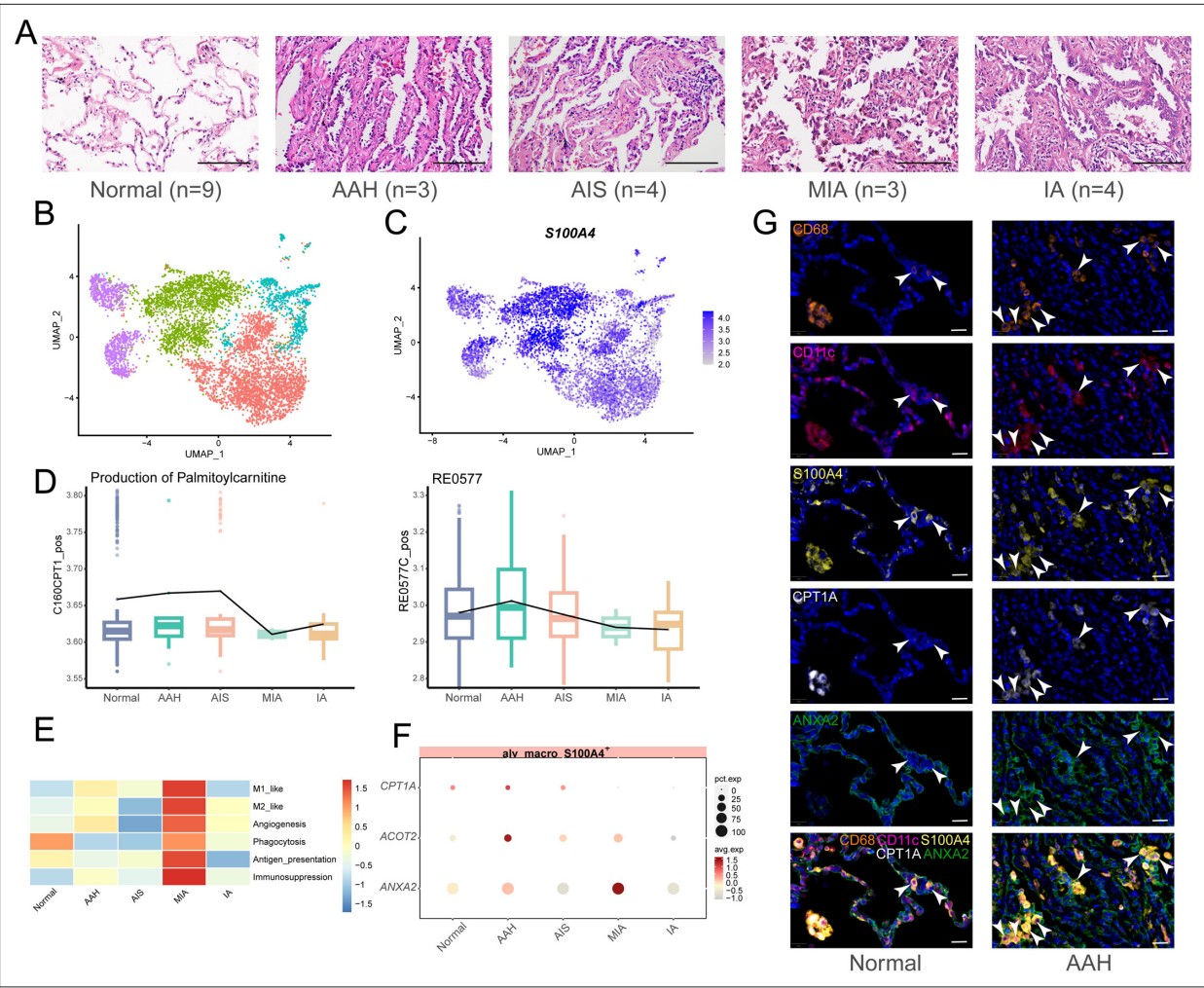

**Figure 7.** *S100A4*[+] alv-macro with a similar pattern in the human AAH stage. (**A**) H&E images of five stages of human lung adenocarcinoma (LUAD) development (normal, adenomatous hyperplasia AAH, adenocarcinoma in situ AIS, minimally invasive adenocarcinoma MIA, and invasive adenocarcinoma IA). Scale bar: 100 μm. (**B**) Uniform Manifold Approximation and Projection (UMAP) plot of subclusters of alveolar macrophages. (**C**) UMAP plot of *S100A4* expression in alveolar macrophages. (**D**) Compass analysis of the reaction activities of fatty acid metabolism in *S100A4*[+] alv-macro across the five stages. (**E**) Macrophage functional analysis of *S100A4*[+] alv-macro across the five stages. (**F**) Dotplot of expression of *CPT1A*, *ACOT2*, and *ANXA2* in the five stages. (**G**) Multiplex immunofluorescence staining of CPT1A and ANXA2 expression in CD68[+]/CD11c[+]/S100A4[+] alv-macro of human normal and AAH tissues. Scale bar: 20 μm.

The online version of this article includes the following figure supplement(s) for figure 7:

**Figure supplement 1.** Metabolic clustering of human single-cell RNA sequencing (scRNA-seq) data and metabolic enrichment analysis of human malignant epithelial cells.

## *S100A4*[+] alv-macro in similar patterns was present in human precancerous AAH lesions

Finally, we were dedicated to corroborating the correspondence between the results analyzed in human samples and our observations in the mouse model. A total of 23 human tissues were included in our study to proceed with the aforementioned analysis, including nine normal lung tissues, three AAHs, four AISs, three MIAs, and four IAs, all of which have been pathologically confirmed by hematoxylin and eosin (H&E) staining and pathologist assessment (*Figure 7A*). The annotation of scRNA-seq profiles yielded 15 cell types, in which macrophages and AT2 cells similarly exhibited significant enrichment of metabolic activities (*Figure 7—figure supplement 1A–C*). Malignant cells were extracted from epithelial cells by inferring chromosomal copy number alterations, and GSVA enrichment analysis of malignant epithelial cells was performed across the four pathological stages (*Figure 7—figure supplement 1D, E*). Along with scMetabolism analysis, the results collectively

suggested that human malignant epithelial cells showed relatively active lipid metabolism during the AAH phase, as for the IA phase, glycometabolism and oxidative phosphorylation pathways were obviously enriched (*Figure 7—figure supplement 1F*). Besides, the enrichment of macrophage migration and chemotaxis indicated the vital role of macrophages throughout the whole malignant transformation process of LUAD.

In order to figure out whether phenotypically and functionally similar subsets of alveolar macrophages also exist in human LUAD precancerous lesions, human alveolar macrophages were further subdivided into four cell subsets, which led us to find that the same was true for human *S100A4*⁺ alv-macro (*Figure 7B and C*). In addition, the activities of palmitoyl-CoA hydrolase and carnitine palmitoyl-transferase were also verified to be elevated during the AAH phase, corresponding to metabolic reactions RE0577C_pos and C160CPT1_pos in the Compass analysis (*Figure 7D*). For the macrophage phenotype, several functional programs with higher abundance in the AAH phase in the mouse scRNA-seq followed a consistent trend in human data, including angiogenesis (*Figure 7E*). In terms of gene expression, the expression of fatty acid metabolism-related genes *CPT1A* and *ACOT2* and angiogenesis-related gene *ANXA2* in the AAH stage was also in line with expectations (*Figure 7F*). Furthermore, we also noticed greater amounts of *S100A4*⁺ alv-macro in human AAH tissues and verified the correlation between CPT1A and ANXA2 in this subset (*Figure 7G*). Taken together, these results confirmed the presence of *S100A4*⁺ alv-macro with fatty acid metabolic properties and pro-angiogenic function in the development of human precancerous AAH.

## Discussion

Here, our scRNA-seq data provided a comprehensive resource to investigate cell state changes alongside tumor initiation in a mouse model of spontaneous tumors simulating the oncogenic transformation in clinical LUAD. We extensively mapped the transcriptional landscape of cell subpopulations and switching cell interactions, some of which were essential in reshaping the microenvironment favoring tumor evolution. In the course of precancerous lesions, the transcriptional heterogeneity was gradually enriched, indicating underlying progressive molecular and cellular changes during carcinogenesis. We identified an initiation-associated marker, *LDHA*; it was suggested that this glycolysis-related gene was significantly upregulated in LUAD and could serve as an independent prognostic indicator of unfavorable overall survival and recurrence-free survival (*Yu et al., 2018*), illustrating the potentially vital role of metabolic alternations in tumorigenesis.

Based on the understanding that TME can affect the state of epithelial cells, metabolic fluctuations in this context are closely linked to cell phenotype and function (*Buck et al., 2017*; *de Visser and Joyce, 2023*; *Wang et al., 2021*). Macrophages are no exception to this rule; their metabolic profile can be shaped by specific histological TME, and the cellular state is thus activated and exhibits remarkable plasticity (*Kumar et al., 2019*; *Mazzone et al., 2018*). Besides, it has been demonstrated that macrophages contribute to the induction of early lung cancer lesions (*Casanova-Acebes et al., 2021*), and purine metabolism has been confirmed to determine the pro-tumor phenotype of the macrophage population (*Li et al., 2022*). In our exploration of malignant epithelial cells in the initiation of LUAD, alveolar macrophages were identified as the cell population with which they interacted most strongly. The scRNA-seq approaches enable the resolution of macrophage heterogeneity, assisting us in understanding how these transcriptionally defined subsets connect to macrophage metabolic status and functional phenotypes. Individual macrophage subsets possess distinct metabolic profiles; our analysis highlighted a specific cell subset, which persisted throughout lung carcinogenesis and was observably defined by the expression of *S100a4* that had previously not been implicated in tumor-associated alveolar macrophages. The most intriguing finding about this cellular state was that the fatty acid metabolism became active during the AAH phase, accompanied by angiogenetic-like properties. S100A4, which was regarded as an angiogenesis regulator, has been stated to enhance macrophage protumor phenotype by control of fatty acid oxidation (*Kazakova et al., 2023*; *Liu et al., 2021*). What is more convincing is that *S100A4*⁺ alv-macro with similar characteristics was extended to scRNA-seq profiles of human precancerous AAH lesions. CPT1A was thought to play a crucial role in this metabolic paradigm shift, and CPT1A-mediated fatty acid oxidation has been proved to be conducive to tumor metastasis, radiation resistance, and immune escape (*Liu et al., 2023*; *Tan et al., 2018*; *Wang et al., 2018*). We demonstrated the regulation of fatty acid metabolism by *Cpt1a* in *S100a4*⁺ alv-macro as well as the involvement of *Pparg*. Nevertheless, the molecular mechanism that

drives the acquisition of metabolic and functional switching properties specific to this cell state still requires further characterization in the context of precancerous lesions. It has been reported that CD36 is the main effector of the S100A4/PPAR-γ pathway, and its mediated fatty acid uptake plays an important role in the tumor-promoting function of macrophages (*Liu et al., 2021*).

In terms of subset functional phenotype, *S100a4*+ alv-macro enriched for fatty acid metabolism held restricted M1-like and phagocytosis capacities at the AAH stage but exhibited M2-like, angiogenesis, and immunosuppressive features, creating a permissive environment for neoplasia. The pro-angiogenic function was dominant and confirmed by tube formation assay of HUVEC, and the regulation of *Cpt1a* on angiogenesis was also revealed. The role of ANXA2 in angiogenesis has been widely recognized, and it may facilitate plasmin production by binding to S100A4 and then trigger angiogenesis and malignant cell transformation (*Grindheim et al., 2017*; *Liu et al., 2015*). Precancerous lesions of LUAD are angiogenic, and pro-angiogenic factors secreted by cells, including *S100a4*+ alv-macro, induce endothelial cell sprouting and chemotaxis, leaving the angiogenic switch activated, prompting the formation of new blood vessels on the basis of the original ones to supply oxygen and nutrients to sustain tumor initiation (*Chen et al., 2025*; *Kayser et al., 2003*; *van Hinsbergh and Koolwijk, 2008*). Under hypoxic conditions, HIF-1α activates numerous factors that contribute to the angiogenic process, including VEGF, which promotes vascular permeability, and MMP9, which breaks down the ECM, promotes endothelial cell migration, and recruits pericytes to provide structural support (*Raza et al., 2010*; *Sakurai and Kudo, 2011*). Cytokines secreted into the microenvironment activate macrophages, which subsequently produce angiogenic factors, further promoting angiogenesis (*Sica et al., 2006*). Furthermore, TGF-β and HGF activate vascular endothelial cells and promote proliferation and migration, as well as induce the expression of pro-angiogenic factors such as VEGF (*Vimalraj, 2022*; *Watabe et al., 2023*). Macrophage-derived TNF-α and IL-1α lead tumor cells to produce potent angiogenic factors IL-8 and VEGF, which affect angiogenesis and tumor growth (*Torisu et al., 2000*). MIP2 and CD40 were also identified as pro-tumor factors associated with angiogenesis (*Kollmar et al., 2006*; *Murugaiyan et al., 2007*). It is worth noting that our mining of *S100a4*+ alv-macro remains at the precancerous initiation stage, and further experimental designs are needed to verify its specific contribution at more aggressive stages. For example, FACS sorting of the subpopulation at different stages of disease progression, respectively, for precise functional characterization; continuous monitoring of the fluctuation of the above factors in bronchoalveolar lavage fluid at corresponding periods; and intratracheal instillation of primary *S100a4*+ alv-macro to observe the pathological progression of precancerous lesions. Besides, as our previous in vitro results were obtained based on cell lines, we will optimize the experimental conditions to achieve coculture of primary macrophage subset and epithelial cells and establish transgenic mouse models for in vivo validation.

With the elaborate resolution of TME, macrophage-related therapy is considered to be promising. So far, macrophage-targeted therapy has demonstrated clinical efficacy in Gaucher's disease and advanced hematological malignancies (*Barton et al., 1991*; *Ossenkoppele et al., 2013*). In lung cancer, an attempt to enhance anti-PD-1 therapy in NSCLC by depleting myeloid-derived suppressor cells with gemcitabine was prematurely terminated because of insufficient data collected; another clinical trial of TQB2928 monoclonal antibody promoting macrophage phagocytosis of tumor cells in combination with a third-generation EGFR TKI for advanced NSCLC is now recruiting. Moreover, preclinical studies on macrophage-targeted therapy combined with immune checkpoint inhibitors are being extensively conducted in NSCLC, and it was suggested that blockade of purine metabolism can reverse macrophage immunosuppression, and a synergetic effect can be achieved when combined with anti-PD-L1 therapy, which inspired the direction of our early intervention strategies (*Wang et al., 2024*; *Yang et al., 2025*).

In this study, our results provided the unique patterns of cellular components during the initiation process of LUAD and gave evidence that *S100a4*+ alv-macro enhances the function of pro-angiogenesis by promoting fatty acid metabolism, thereby accelerating the process of precancerous lesions (*Figure 8*). Early intervention in solid cancers at the precancer stage may effectively prevent their progression (*Prieto et al., 2023*). In the cascade of tumor initiation, AAH is represented as a probable forerunner of LUAD, serving as a potential hub in carcinogenesis (*Mori et al., 2001*). On the basis of the newly identified pro-tumor subpopulation and relevant metabolic and functional elements, the development of novel immunotherapies targeting the immunometabolism will provide alternative choices for interventions at the AAH stage, paving the way for potential benefits for patients.

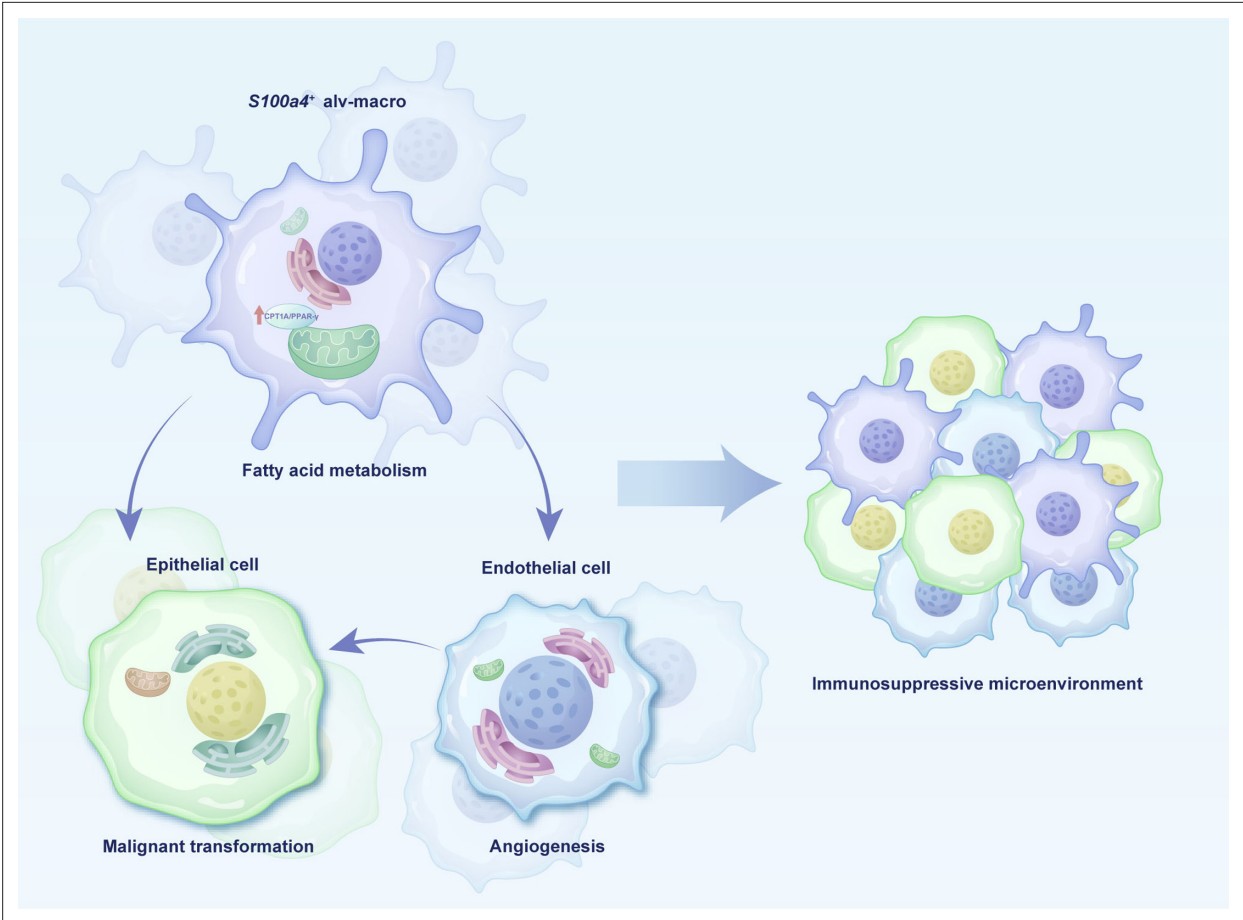

**Figure 8.** Schema of cell interactions in the precancerous microenvironment.

## Materials and methods

### A/J mouse model and human samples

A/J mice were obtained from the Shanghai Model Organisms Center (Shanghai, China). The 200 mice were divided into groups and housed in an air-conditioned room with a 12 hr light/dark cycle and standard chow diet. Mice were killed by decapitation at different months of age. A gross examination of all organs was conducted, the lungs were removed and bisected for tissue fixation and the preparation of single-cell suspensions. All the animal surgical and experimental procedures were approved by the Ethics Board of Experimental Animals, West China Hospital, Sichuan University (Approval No. 20230208003).

Human clinical specimens from ten patients were enrolled with informed consent from West China Hospital. All samples were obtained during surgery and prior to any other treatment. Histological classification of precancerous lesions and tumors was determined according to the WHO guidelines (*Travis et al., 2015*). Fresh specimens were instantly divided into two pieces for single-cell suspension preparation and histopathological staining. The utilization of human samples was approved by the Medical Ethics Committee and Institutional Review Board of West China Hospital, with all participants providing written informed consents.

### H&E staining

Lung tissue was further analyzed by histopathology to confirm pathological stage. H&E staining was performed on 4 µm paraffin sections according to routine staining protocol. Microscopic H&E-stained slides were independently reviewed by clinical or veterinary pathologists from the Department of Pathology, West China Hospital. Histopathological classification of mouse (AAH, adenoma, and AIS)

and human (AAH, AIS, MIA, and IA) lesions was performed as previously described (*Marjanovic et al., 2020*; *Sivakumar et al., 2017*; *Travis et al., 2011*; *Weichert and Warth, 2014*).

## Single-cell suspension preparation

Resected samples were washed in Hanks' balanced salt solution (HBSS, GIBCO, Grand Island, USA) on ice immediately, minced into small pieces with a sterile surgical scalpel, and then transferred into a conical tube. The mechanical dissociation was followed by an enzymatic digestion. The tissue fragments were digested in HBSS with 2 mg/mL collagenase I and 1 mg/mL collagenase IV (Washington, Lakewood, USA) for 30–45 min at 37 °C. The suspension was then filtered using 70 μm and 30 μm cell strainers (Corning, NY, USA) successively, and the cell pellet was resuspended with red blood cell (RBC) lysis buffer (Invitrogen, Carlsbad, USA) to discard RBCs. Following cell sorting, dead cells were removed from the cell suspension based on exclusion of 7-aminoactinomycin D (7-AAD, Life Technologies Corporation, Gaithersburg, USA), and live cells were sorted into phosphate buffer solution (PBS) with 0.04% bovine serum albumin (BSA) for subsequent scRNA-seq.

## Single-cell RNA library construction and sequencing

Single-cell capturing and cDNA library generation were performed using the Single Cell 3' Chip Kit and 10 x Chromium 3' Library & Gel Bead Kit v3 (10 x Genomics, Pleasanton, USA) following the manufacturer's instructions. The barcoded cDNA was purified with Dynabeads MyOne SILANE beads (Thermo Fisher Scientific, Waltham, USA) before amplification by PCR. Amplified cDNA product was size-selected using SPRIselect Reagent (Thermo Fisher Scientific). After adaptor ligation and sample index PCR, libraries were sequenced on an Illumina NovaSeq 6000.

## Single-cell RNA sequencing analysis and identification of marker genes

Raw gene expression matrices were generated per sample using CellRanger (version 4.0.0). These matrices were then combined by R (version 4.0.5) and converted into a Seurat data object using the Seurat R package (version 4.3.0.1). We excluded cells based on any one of the following criteria: (1) more than 10,000 or fewer than 500 unique molecular identifiers (UMIs); (2) more than 5000 or fewer than 300 expressed genes; (3) over 10% UMIs that were derived from the mitochondrial genome; (4) genes that were expressed in fewer than 10 cells. Subsequently, we carried out downstream analyses by the Seurat workflow (*Hao et al., 2021*) as described below:

First, we normalized the counts on the filtered matrix for each gene to the total library size by the 'NormalizeData' function. Second, we employed the 'FindVariableGenes' function to identify the most highly variable genes for unsupervised clustering. Third, these highly variable genes were centered to a mean of zero and were scaled by the standard deviation by the 'ScaleData' function of Seurat. Fourthly, principal component analysis (PCA) was performed by the function 'RunPCA.' The 'RunHarmony' function was used to integrate samples and remove batch effects. The 'FindClusters' function was used to calculate the clusters of the cells. UMAP plots were used to visualize clusters of cells localized in the graph-based clusters by the 'RunUMAP' function. Markers for each cluster were identified by finding differentially expressed genes between cells from the individual cluster versus cells from all other clusters by the 'FindAllMarkers' function.

## Estimating heterogeneity within stages

Heterogeneity of single-cell profiles within a stage was quantified by examining the average pairwise Normalized Mutual Information (NMI) (*Marjanovic et al., 2020*) between the profiles for each stage. First, we selected 100 differentially expressed genes for each of the 13 cell type clusters and the top 100 genes for each of the top 20 components in harmony reduction. Next, we discretized expression per gene into 10 bins. Considering the differences in the number of cells across samples, we subsampled 100 cells from each stage 500 times and calculated the median NMI across each within-stage sampled pair. NMI was calculated between each pair of cells x and y by first calculating the mutual information (*Equation 1*) and then normalizing it by the entropy of each cell (*Equation 2*).

$$I\left(x,y\right) = \sum_x \sum_y p\left(x,y\right) log \frac{p(x,y)}{p(x)p(y)} \tag{1}$$

$$NMI\left(x, y\right) = \frac{I(x,y)}{\sqrt{H(x)H(y)}}. \tag{2}$$

where $p\left(x\right)$ and $p\left(y\right)$ represent the probability distributions of cell x and cell y over all discretized gene expression values, respectively. $p\left(x, y\right)$ denotes the joint probability distribution of cell x and cell y over all discretized gene expression values. $H\left(x\right)$ and $H\left(y\right)$ correspondingly indicate the entropy of cell x and cell y across all discretized gene expression values.

## Single-cell metabolic state analysis

According to the Molecular Signatures Database (*Subramanian et al., 2005*), we extracted 2710 genes associated with the metabolic pathways for dimensional reduction and clustering analysis. We employed PCA to process the metabolic genes and used UMAP to do dimensional reduction for the top 20 components. After that, we employ the 'FindClusters' function of Seurat for clustering analysis. Next, we used the 'FindAllMarkers' function of Seurat to detect the specific expression of metabolic genes for each metabolic state and used Single-sample GSEA to explore the dynamic pattern of metabolic pathways by SingleSeqGset (*Cillo, 2021*) (version 0.1.2.9) of R.

## Pathway activity analysis

GSVA (*Hänzelmann et al., 2013*) is a widely used approach to assess pathway enrichment of samples for gene expression data. We used the GSVA package (version 1.51.0) to carry out enrichment analysis for the annotated KEGG and Gene Ontology_Biological Process (GOBP) gene sets in the msigdbr package (*Subramanian et al., 2005*) (version 7.5.1), followed by differential pathway analysis using the limma package (*Ritchie et al., 2015*) (version 3.54.2). We not only visualized such pathways that have significant differences (adj. p.val <0.05), but also employed the scMetabolism package (*Wu et al., 2022*) (version 0.2.1) to do enrichment analysis and visualization for KEGG and Reactome metabolic pathways.

## Cell-cell communication analysis

We employed the CellChat (version 1.6.1) algorithm (*Jin et al., 2021*) to analyze and quantify cell-cell communication between all major cell types. CellChat identifies differentially over-expressed ligands and receptors for each cell group and models the probability of communication between cell groups by the law of mass action. Significant interactions are identified based on a permutation test (*Good, 2013*) which can randomly permute the group labels of cells. We then used chord plots and circle plots to describe the number of cell-cell interactions and communication strength and visualized significant interactions (ligand-receptor pairs) from some cell groups to other cell groups by the 'netVisual_bubble' function of CellChat.

## InferCNV analysis

Malignant epithelial cells were identified using inferCNV (*Tickle et al., 2019*) (version 1.14.2). We scored each cell for the extent of CNV signal and plotted the cells on a dendrogram, and then we cut the dendrogram into 6 clusters. We labeled all cells clustered with normal cells in the control group as 'non-mag' and the remaining clusters as 'mag'.

## Correlation analysis between cell groups

We calculated the correlation between different cell groups based on the expression of each cell on the genes. First, the mean value of gene expression was taken as the representative value for each group of cells on each gene. Then, the top 1000 genes with the greatest standard deviation were extracted to calculate the correlation between groups by *Spearman, 1987* method, and finally we used the corrplot (*Wei et al., 2013*) (version 0.92) function to visualize the correlation matrix.

## Cellular metabolic activity modeling

We modeled the metabolic state of individual cells by the Compass algorithm (*Wagner et al., 2021*). We extracted the normalized gene expression matrix as input and then ran Compass with default settings. The output of Compass is a reactions.tsv file, where the numbers correspond to penalties for

each reaction per cell. We employ a negative log to have scores. If the score is great, the reaction is more active.

## Gene set module score

The AddModuleScore function of Seurat can be used to score any gene set. Its workflow is to first calculate the average value of all genes in the target gene set on a single-cell level, second divide the expression matrix into several bins based on the average value, and third randomly select the control genes (genes outside the target gene set, default 100 genes) as the background value from each bin. Finally, we count all target and background genes as an average value, respectively. The score for the gene set is determined by subtracting the average background value from the average target gene value. Here, we used this function to score gene sets of interest at the single-cell level.

## Correlation analysis between gene sets

For correlation analysis between gene sets, the score of each gene set was defined as above. Here, we used the (*Pearson, 1901*) method for correlation analysis and employed the ggpubr package (*Kassambara, 2020*) (version 0.6.0) to plot scatter plots.

## Macrophage function analysis

Gene sets for well-known macrophage functions (M1-like polarization, M2-like polarization, Angiogenesis, Phagocytosis, Antigen presentation, and Immunosuppression) were collected from databases and relevant literature (*Supplementary file 1*). The GSVA package was used to analyze the functional enrichment of these gene sets.

## Immunofluorescence staining

The sections of mouse lung tissue were first blocked with normal goat serum (10%). Primary antibodies were diluted and incubated with the sections at 4 °C overnight: anti-Ki67 (Abcam, Cambridge, USA) and anti-LDHA (Proteintech, Wuhan, China). The slides were then washed three times with PBS and incubated with Alexa-conjugated secondary antibodies (Jackson ImmunoResearch Laboratories, West Grove, USA) for 1 hr at room temperature. The nuclei were finally counterstained with DAPI (4', 6-diamidino-2-phenylindole, Sigma-Aldrich, St. Louis, USA), and the sections were mounted. Images were captured by a fluorescence microscope (Olympus, Tokyo, Japan). The pictures presented represent the results of three sections, each of which was collected for nine fields.

## Multiplex immunohistochemistry staining

Multiplex immunohistochemistry staining is a tyramine amplification-based method for in situ labeling of tissue proteins. Combined with spectroscopic imaging technology, the rich protein information contained in the tissue can be obtained. The New Opal Polaris 7-color Manual IHC Kit (Akoya Biosciences, Massachusetts, USA) was used for multiple staining. The Opal multi-labeling method used indirect fluorescent labeling in which tissues were incubated with primary antibody and then labeled with horseradish peroxidase (HRP) secondary antibody and fluorescence amplification working solution. The non-covalently bound antibody was then washed by thermal repair, then replaced with another primary antibody and fluorescein substrate, and so on to achieve multiple labeling, followed by DAPI staining and anti-fluorescence quencher sealing. Detailed experimental procedures were described in the kit instructions. The multispectral tissue imaging system (PerkinElmer Vectra) and analysis software (Phenochart 1.0) were used for image processing. The following antibodies were used in the staining: F4/80 (Cell Signaling Technology, Danvers, USA), CD11c (Abcam), S100A4 (Abcam), CPT1A (Abcam), ANXA2 (Abcam), and CD68 (Abcam).

## Transient transfection in MH-S cells

The murine alveolar macrophage MH-S cell line was purchased from Procell Life Science & Technology (Wuhan, China) and cultured in RPMI-1640 medium (GIBCO) supplemented with 10% fetal bovine serum (FBS, GIBCO) and 0.05 mM 2-mercaptoethanol (Sigma-Aldrich). The plasmids of *S100a4* and NC were commercially obtained from Hanbio (Shanghai, China) and transiently transfected into MH-S cells with Lipofectamine 3000 (Invitrogen, Carlsbad, CA), respectively, according to the manufacturer's

instructions. *S100a4* siRNAs (OBiO, Shanghai, China) were also transfected in the manner described above. ETO (MedChem Express) was used for targeted inhibition of CPT1A in MH-S.

## Coculture assay in vitro

The mouse alveolar epithelial MLE12 cell line was kindly provided by Dr. Yi Li and cultured in DMEM/F12 medium (GIBCO) with 10% FBS. For the coculture experiment, the transfected MH-S cells were incubated with serum-free medium overnight. The CM was centrifuged at 200 g for 5 min and filtered through a 0.22 μm syringe filter. The collected CM was then applied to MLE12 cells.

## qPCR

Total RNA was extracted by the SevenFast Total RNA Extraction Kit (Seven, Beijing, China). Reverse transcription for cDNA conversion was conducted with PrimeScript RT Master Mix (TAKARA, Tokyo, Japan). qPCR analysis of mRNA expression was measured by TB Green Premix Ex Taq (TAKARA) and normalized to β-actin. All reactions were performed in triplicate. The primers for target genes are listed below:

> *S100a4*-F: TCCACAAATACTCAGGCAAAGAG
> *S100a4*-R: GCAGCTCCCTGGTCAGTAG
> *β-actin*-F: GGCTGTATTCCCCTCCATCG
> *β-actin*-R: CCAGTTGGTAACAATGCCATGT

## Western blotting

Total protein was extracted using lysis buffer containing protease inhibitor cocktail (MedChem Express, Monmouth Junction, USA) and quantitated by BCA protein assay kit (KeyGEN BioTECH, Beijing, China). Proteins were separated by sodium dodecyl sulfate-polyacrylamide gel electrophoresis (SDS-PAGE) and then transferred to a polyvinylidene fluoride (PVDF) membrane. Immunoblotting was performed with the following primary antibodies: S100A4 (Abcam), β-Actin (Cell Signaling Technology), SPA (Boster, Wuhan, China), E-Cadherin (Proteintech), N-Cadherin (ABclonal, Wuhan, China), Vimentin (ABclonal), CD44 (Affinity, Jiangsu, China), CD133 (Proteintech), p-ERK (Santa Cruz Biotechnology, Santa Cruz, USA), MMP9 (Cell Signaling Technology), p-γH2AX (Abcam), VEGF (Santa Cruz Biotechnology), TGF-β (Cell Signaling Technology), HIF-1α (Abcam), c-MYC (Abcam), CPT1A (Abcam), ANXA2 (Abcam), PLG (Abcam), PPAR-γ (Proteintech), CD206 (Abcam), and ARG1 (Proteintech).

## Cell proliferation and migration assay

Cell viability was assessed by using Cell Counting Kit-8 (CCK8, 4 A Biotech Co., Beijing, China). Cell proliferation was measured by colony formation assay. Cell migration was determined by a wound-healing assay.

## Cell cycle assay

PI/RNase Staining Solution (Sungene Biotech Co., Tianjin, China) was used for cell cycle distribution measurement.

## Intracellular ROS measurement

The intracellular ROS level was detected using the Reactive Oxygen Species Assay Kit (Beyotime, Shanghai, China). After coculture, MLE12 cells were loaded with DCFH-DA probe in situ and incubated in a 37 °C incubator for 30 min. Cells were harvested, and the fluorescence intensity was measured by flow cytometry.

## Transmission electron microscopy

Cell samples were prefixed with 2.5% glutaraldehyde, refixed with 1% osmium tetroxide, dehydrated step by step with acetone, and then subjected to osmosis and embedded with Epon-812 embedding agent. 60–90 nm ultrathin sections were made by an ultrathin microtome and transferred to copper mesh, then stained with uranyl acetate for 10–15 min, followed by lead citrate for 1–2 min at room temperature. Image acquisition was carried out with a JEM-1400FLASH transmission electron microscope (JEOL, Japan).

## Mouse cytokine antibody array

CM from *S100a4*-OE and NC MH-S were collected for cytokine detection by RayBio G-Series Arrays (Guangzhou, China). Like a traditional sandwich-based ELISA, this array uses a matched pair of cytokine-specific antibodies for detection. After incubation with the sample, the target cytokines are captured by the antibodies printed on the solid surface. A second biotin-labeled detection antibody is then added, which recognizes a different epitope of the target cytokine. The cytokine-antibody-biotin complex can then be visualized through the addition of the streptavidin-conjugated Cy3 equivalent dye. After capturing the spot densities with a laser scanner, array data analysis, and functional enrichment analysis were conducted.

## Measurement of lipid droplet content and mitochondrial function

BODIPY 493/503 (GLPBIO, Montclair, USA) was used to detect intracellular lipid droplet accumulation. TMRM and MitoSOX Green (Thermo Fisher Scientific) were used to assess mitochondrial membrane potential and mitochondrial ROS production. The fluorescence intensity was measured by flow cytometry.

## Tube formation assay

HUVECs ($2\times10^4$ cells per well) were seeded in 96-well plates coated with Matrigel (Corning) and incubated in the CM from MH-S. Tube formation was examined under an inverted microscope after 6 hr. The total tube length and the number of nodes were measured using ImageJ software.

## Statistics

Statistical significance was determined using a two-tailed unpaired Student's t-test, and p-value <0.05 was considered significant (*p<0.05, **p<0.01, ***p<0.001, ****p<0.0001). The correlation coefficient (R) was calculated using the Pearson method with 95% confidence intervals (CI), and corresponding p-values were added to the plots. Statistical analyses were performed using R, Python, and GraphPad Prism.

## Acknowledgements

We thank Home for Researchers (https://www.home-for-researchers.com/) for the assistance in the professional optimization of the schematic diagram. This work was supported by the National Natural Science Foundation of China [92159302 and 62372316], National Science and Technology Major Project [2021YFF1201200 and 2024ZD0532900], Sichuan Science and Technology Program Key Project [2024YFHZ0091], Sichuan Science and Technology Project [2022ZDZX0018], and 1.3.5 Project for Disciplines of Excellence, West China Hospital, Sichuan University [ZYGD2200].

## Additional information

### Funding

| Funder | Grant reference number | Author |
|---|---|---|
| National Natural Science Foundation of China | 92159302 | Li Zhang |
| National Natural Science Foundation of China | 62372316 | Le Zhang |
| National Science and Technology Major Project | 2021YFF1201200 | Le Zhang |
| Sichuan Science and Technology Program Key Project | 2024YFHZ0091 | Le Zhang |
| Sichuan Science and Technology Project | 2022ZDZX0018 | Li Zhang |

| Funder | Grant reference number | Author |
|---|---|---|
| West China Hospital, Sichuan University | ZYGD2200 | Li Zhang |
| National Science and Technology Major Project | 2024ZD0532900 | Le Zhang |

The funders had no role in study design, data collection and interpretation, or the decision to submit the work for publication.

## Author contributions

Hong Huang, Conceptualization, Resources, Validation, Investigation, Visualization, Methodology, Writing – original draft, Writing – review and editing; Ying Yang, Resources, Validation, Investigation, Methodology; Qiuju Zhang, Data curation, Software, Formal analysis, Visualization; Yongfeng Yang, Conceptualization, Methodology, Writing – review and editing; Zhenqi Xiong, Data curation, Software, Formal analysis; Shengqiang Mao, Tingting Song, Software, Formal analysis; Yilong Wang, Validation, Investigation; Zhiqiang Liu, Resources, Validation; Hong Bu, Conceptualization, Supervision, Project administration, Writing – review and editing; Li Zhang, Conceptualization, Resources, Supervision, Funding acquisition, Project administration, Writing – review and editing; Le Zhang, Software, Supervision, Funding acquisition, Project administration, Writing – review and editing

## Author ORCIDs

Hong Huang https://orcid.org/0000-0002-2706-1570
Ying Yang https://orcid.org/0000-0002-1871-3119
Qiuju Zhang https://orcid.org/0009-0000-9454-6650
Yongfeng Yang https://orcid.org/0000-0002-8109-9801
Zhenqi Xiong https://orcid.org/0009-0005-6128-9965
Shengqiang Mao https://orcid.org/0000-0001-5431-2227
Tingting Song https://orcid.org/0000-0001-8965-3793
Yilong Wang https://orcid.org/0009-0007-8130-8857
Zhiqiang Liu https://orcid.org/0000-0002-7684-968X
Hong Bu https://orcid.org/0000-0001-7472-6443
Li Zhang https://orcid.org/0000-0003-1125-1991
Le Zhang https://orcid.org/0000-0002-3708-1727

## Ethics

The utilization of human samples was approved by the Medical Ethics Committee and Institutional Review Board of West China Hospital, with all participants providing written informed consents.
All the animal surgical and experimental procedures were approved by the Ethics Board of Experimental Animals, West China Hospital, Sichuan University (Approval No. 20230208003).

Reviewer #1 (Public review): https://doi.org/10.7554/eLife.101731.3.sa1
Reviewer #2 (Public review): https://doi.org/10.7554/eLife.101731.3.sa2
Author response https://doi.org/10.7554/eLife.101731.3.sa3

---

# Additional files

## Supplementary files

Supplementary file 1. List of genes for macrophage functions.

MDAR checklist

## Data availability

The scRNA-seq data of mouse and human have been deposited to Genome Sequence Archive (https://ngdc.cncb.ac.cn/gsa/browse/CRA025954) and Genome Sequence Archive for Human (https://ngdc.cncb.ac.cn/gsa-human/browse/HRA011594), respectively. The image data have been deposited to BioImage Archive (https://doi.org/10.6019/S-BIAD1943).

The following datasets were generated:

| Author(s) | Year | Dataset title | Dataset URL | Database and Identifier |
|---|---|---|---|---|
| Zhang L | 2025 | S100a4+ alveolar macrophages accelerate the progression of precancerous atypical adenomatous hyperplasia by promoting the angiogenic function regulated by fatty acid metabolism | https://ngdc.cncb.ac.cn/gsa/browse/CRA025954 | Genome Sequence Archive, CRA025954 |
| Zhang L | 2025 | S100a4+ alveolar macrophages accelerate the progression of precancerous atypical adenomatous hyperplasia by promoting the angiogenic function regulated by fatty acid metabolism. | https://ngdc.cncb.ac.cn/gsa-human/browse/HRA011594 | Genome Sequence Archive for Human, HRA011594 |
| Zhang L | 2025 | S100a4+ alveolar macrophages accelerate the progression of precancerous atypical adenomatous hyperplasia by promoting fatty acid metabolism | https://doi.org/10.6019/S-BIAD1943 | BioImage Archive, 10.6019/S-BIAD1943 |

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
