## [Editor Report · eLife Assessment]

This is an **important** study demonstrating the importance of S100A4+ alveolar macrophages in the earlier stages of tumour development and suggesting a role in angiogenesis. As such this **convincing** study is of interest to cancer biologists focused on early tumour development and those interested in the development of therapeutics that may specifically target early cancers.

---

## [Referee Report · Reviewer #1 (Public review)]

Summary:

In this paper, the authors have leveraged Single-cell RNA sequencing of the various stages of evolution of lung adenocarcinoma to identify the population of macrophages that contribute to tumor progression. They show that S100a4+ alveolar macrophages, active in fatty acid metabolic activity, such as palmitic acid metabolism, seem to drive atypical adenomatous hyperplasia (AAH) stage. These macrophages also seem to induce angiogenesis promoting tumor growth. Similar types of macrophage infiltration were demonstrated in the progression of the human lung adenocarcinomas.

Comments on revised version:

The authors have satisfactorily addressed my main concerns.

The only weakness is that infusion of S100a4+ macrophages seem not to affect tumor growth when introduced to the intratracheal route. This negative result somewhat diminishes the significance of the study.

Overall, the revised manuscript is significantly improved.

---

## [Referee Report · Reviewer #2 (Public review)]

Summary:

The work aims to further understand the role of macrophages in lung precancer/lung cancer evolution

Strengths:

(1) The use of single-cell RNA seq to provide comprehensive characterisation.

(2) Characterisation of cross-talk between macrophages and the lung precancerous cells.

(3) Functional validation of the effects of S100a4+ cells on lung precancerous cells using in vitro assays.

(4) Validation in human tissue samples of lung precancer / invasive lesions.

Weaknesses identified previously:

(1) The authors need to provide clarification of several points in the text.

(2) The authors need to carefully assess their assumptions regarding the role of macrophages in angiogenesis in precancerous lesions.

(3) The authors should discuss more broadly the current state of anti-macrophage therapies in the clinic.

Comments on revised version:

The authors have adequately addressed all of my comments.

---

## [Author Response]

The following is the authors’ response to the original reviews

**Public Reviews:**

**Reviewer #1 (Public review):**
Summary:In this paper, the authors have leveraged Single-cell RNA sequencing of the various stages of the evolution of lung adenocarcinoma to identify the population of macrophages that contribute to tumor progression. They show that S100a4+ alveolar macrophages, active in fatty acid metabolic activity, such as palmitic acid metabolism, seem to drive the atypical adenomatous hyperplasia (AAH) stage. These macrophages also seem to induce angiogenesis promoting tumor growth. Similar types of macrophage infiltration were demonstrated in the progression of the human lung adenocarcinomas.Strengths:Identification of the metabolic pathways that promote angiogenesis-dependent progression of lung adenocarcinomas from early atypical changes to aggressive invasive phenotype could lead to the development of strategies to abort tumor progression.

We are grateful for your constructive comments. These comments are very helpful for revising and improving our paper and have provided important guiding significance to our study. We have made revisions according to your comments and have provided point-by-point responses to your concerns.

Weaknesses:(1) Can the authors demonstrate what are the functional specialization of the S100a4+ alveolar macrophages that promote the progression of the AAH to the more aggressive phenotype? What are the factors produced by these unique macrophages that induce tumor progression and invasiveness?

Thank you for your comments. To more comprehensively characterize the functional specialization of the S100a4^+^ alveolar macrophages, we expanded the macrophage functional gene sets based on relevant literature and databases and performed enrichment analysis. The results showed that all stages of precancerous progression presented activated states of angiogenesis, M2-like and immunosuppressive functions relative to the normal stage (Figure 4B). As we have demonstrated, S100a4^+^ alveolar macrophages predominantly exert pro-angiogenic functions during the AAH phase and may be more biased towards M2-like polarization and immunosuppression during further disease progression. Consistently, S100A4^+^ subset population of macrophages has been proved to exhibit a M2-like phenotype with immunosuppressive properties in tumor progression [PMID: 34145030]. In addition, S100A4 has been reported to be associated with macrophage M2 polarization, angiogenesis, and tumorigenesis [PMID: 39664586, 36895491, 30221056, 32117590]. The functional status of human S100A4^+^ alveolar macrophages is basically the same. The relevant description was added to the Results section as follows: “It was revealed that the capacities for angiogenesis, M2-like polarization, and immunosuppression were found to be stronger in AAH or other precancerous stages relative to the normal stage (Figure 4B). The pro-angiogenic function predominated in the AAH stage, while M2-like and immunosuppressive functions were more prominent in the subsequent precancerous progression.” (page 11, line 262). Our study puts more attention on the functional phenotypic changes of S100a4^+^ alveolar macrophages during the progression from normal to AAH to explain the role of this subpopulation in tumor initiation, and similarly, preliminary coculture experiments could only indicate its role in the early malignant transformation of epithelial cells. In further experimental validation, we will confirm the above functions of the S100a4^+^ alveolar macrophages promoting the progression of AAH to the more aggressive phenotype by in vitro and in vivo experiments. We have extended the limitations and potential experimental designs to the Discussion section as follows: “It is worth noting that our mining of S100a4^+^ alv-macro remains at the precancerous initiation stage, and further experimental designs are needed to verify its specific contribution at more aggressive stages. For example, FACS sorting of the subpopulation at different stages of disease progression, respectively, for precise functional characterization;” (page 19, line 468).

For the factors produced by these unique macrophages during induction of malignant transformation, we assayed culture supernatant of S100a4-OE alveolar macrophages for secreted functional cytokines. The results showed up-regulation of MIP-2, HGF, TNFα, IL-1a, CD27, CT-1, MMP9, 4-1BB, and CD40, and GO enrichment showed angiogenesis and tumorigenesis-related processes (Figure 5L and 5M). We have added the detailed content to the Results section as follows: “Next, we detected tumor-inducing factors secreted by these unique macrophages using Cytokine Antibody Array. We noted the production of macrophage inflammatory protein (MIP)-2, hepatocyte growth factor (HGF), tumor necrosis factor α (TNF-α), IL-1α, MMP9, and CD40, and these cytokine-related biological processes were mainly involved in the regulation of angiogenesis and immune response (Figure 5L and 5M).” (page 13, line 319). Furthermore, changes in these cytokines during subsequent invasive tumor progression will also be continuously monitored. The description in the Discussion section have been added as: “Furthermore, TGF-β and HGF activate vascular endothelial cells and promote proliferation and migration, as well as induce the expression of pro-angiogenic factors such as VEGF (Vimalraj, 2022; Watabe, Takahashi, Pietras, & Yoshimatsu, 2023). Macrophage-derived TNF-α and IL-1α lead tumor cells to produce potent angiogenic factors IL-8 and VEGF, which affect angiogenesis and tumor growth (Torisu et al., 2000). MIP2 and CD40 were also identified as pro-tumor factors associated with angiogenesis (Kollmar, Scheuer, Menger, & Schilling, 2006; Murugaiyan, Martin, & Saha, 2007)…continuous monitoring of the fluctuation of the above factors in bronchoalveolar lavage fluid at corresponding periods;” (page 19, line 461).

All method details covered in this section have been updated in the Materials and methods.

(2) Angiogenic factors are not only produced by the S100a4+ cells but also by pericytes and potentially by the tumor cells themselves. Then, how do these factors aberrantly trigger tumor angiogenesis that drives tumor growth?

Thank you for your comment. In our study, we detected up-regulation of angiogenic factors HIF-1α, VEGF, MMP9, and TGF-β (Figure 5K), and elevation of secreted HGF, IL-1α, and TNF-α (Figure 5L). We provide a detailed description of how these factors are involved in angiogenesis-related tumorigenesis to varying degrees in the Discussion section: “Precancerous lesions of LUAD are angiogenic, and pro-angiogenic factors secreted by cells, including S100a4^+^ alv-macro, induce endothelial cell sprouting and chemotaxis, leaving the angiogenic switch activated, prompting the formation of new blood vessels on the basis of the original ones to supply oxygen and nutrients to sustain tumor initiation (Chen et al., 2024; Kayser et al., 2003; van Hinsbergh & Koolwijk, 2008). Under hypoxic conditions, HIF-1α activates numerous factors that contribute to the angiogenic process, including VEGF, which promotes vascular permeability, and MMP9, which breaks down the ECM, promotes endothelial cell migration, and recruits pericytes to provide structural support (Raza, Franklin, & Dudek, 2010; Sakurai & Kudo, 2011). Cytokines secreted into the microenvironment activate macrophages, which subsequently produce angiogenic factors, further promoting angiogenesis (Sica, Schioppa, Mantovani, & Allavena, 2006). Furthermore, TGF-β and HGF activate vascular endothelial cells and promote proliferation and migration, as well as induce the expression of pro-angiogenic factors such as VEGF (Vimalraj, 2022; Watabe, Takahashi, Pietras, & Yoshimatsu, 2023). Macrophage-derived TNF-α and IL-1α lead tumor cells to produce potent angiogenic factors IL-8 and VEGF, which affect angiogenesis and tumor growth (Torisu et al., 2000)…” (page 19, line 449).

(3) It is not clear how abnormal fatty acid uptake by the macrophages drives the progression of tumors.

Thank you for your comment, which coincides with our mechanistic exploration. The metabolic status of macrophages influences their pro-tumor properties, and lipid metabolism has been shown to determine the functional polarization of macrophages [PMID: 29111350]. In this study, we observed more accumulation of lipid droplets in S100a4-OE MH-S, demonstrating enhanced cellular fatty acid uptake (Figure 6A). The pro-angiogenic ability of S100a4^+^ alv-macro was confirmed by tube formation assay and cytokine assay (Figure 6B and 5M). Cpt1a was thought to play a crucial role in the metabolic paradigm shift of S100a4^+^ alv-macro, we therefore performed functional rescue experiments by inhibiting CPT1A expression in S100a4-OE MH-S by addition of etomoxir (ETO). After culture with conditioned medium of MH-S, the proliferation, migration, and ROS production of MLE12 cells were all restored to lower levels (Figure 6E-G). In addition, ETO treatment significantly reversed the angiogenesis, which supported the regulation of fatty acid metabolism on macrophage function (Figure 6H). Immunoblotting also revealed restoration of expression in related proteins (Figure 6I and 6J), these findings reinforced previous analyses of the association of fatty acid metabolism with pro-angiogenesis and M2-like function in S100a4^+^ alv-macro. The involvement of PPAR-γ in the regulation of metabolic state was also confirmed. Taken together, we suggest that S100a4^+^ alv-macro promotes fatty acid metabolism through the CPT1A-PPAR-γ axis, enhances its ability to promote angiogenesis, and thus drives tumor occurrence. The corresponding contents were added in the Results section S100a4^+^ alv-macro drove angiogenesis by promoting Cpt1a-mediated fatty acid metabolism (page 13, line 327) and Discussion section: “We demonstrated the regulation of fatty acid metabolism by CPT1A in S100a4^+^ alv-macro as well as the involvement of PPAR-γ. Nevertheless, the molecular mechanism that drives the acquisition of metabolic and functional switching properties specific to this cell state still requires further characterization in the context of precancerous lesions. It has been reported that CD36 is the main effector of the S100A4/PPAR-γ pathway, and its mediated fatty acid uptake plays an important role in the tumor-promoting function of macrophages (S. Liu et al., 2021).” (page 18, line 433).

All method details covered in this section have been supplemented in the Materials and methods.

(4) Does infusion or introduction of S100a4+ polarized macrophages promote the progression of AAH to a more aggressive phenotype?

Thank you for your comment. We performed intratracheal instillation of lentivirus-infected S100a4-OE MH-S and culture supernatant in A/J and BALB/c mice, respectively, but no aggressive pathological phenotype was observed so far, possibly due to the lack of time required for lesions or the imperfection of experimental conditions. We will continue to explore the instillation dose and frequency for long-term monitoring and will simultaneously evaluate the availability of primary alveolar macrophages. We have discussed as follows: “It is worth noting that our mining of S100a4^+^ alv-macro remains at the precancerous initiation stage, and further experimental designs are needed to verify its specific contribution at more aggressive stages…and intratracheal instillation of primary S100a4^+^ alv-macro to observe the pathological progression of precancerous lesions.” (page 19, line 468).

(5) How does Anxa and Ramp1 induction in inflammatory cells induce angiogenesis and tumor progression?

Thank you for your comment. ANXA2 is an important member of annexin family of proteins expressed on surface of endothelial cells, macrophages, and tumor cells [PMID: 30125343]. ANXA2 was reported to regulate neoangiogenesis in the tumor microenvironment and most likely due to overproduction of plasmin. As a well-established receptor for plasminogen (PLG) and tissue plasminogen activator (tPA) on the cell surface, ANXA2 converts PLG into plasmin. Plasmin plays a critical role in the activation of cascade of inactive proteolytic enzymes such as metalloproteases (pro-MMPs) and latent growth factors (VEGF and bFGF) [PMID: 12963694, 11487021]. Activated forms of MMPs and VEGF then induce extracellular matrix remodeling facilitating angiogenesis and tumor development [PMID: 15788416]. Sharma et al. suggested administration of ANXA2-antibody inhibited tumor angiogenesis and growth concurrent with plasmin generation [PMID: 22044461], the role of ANXA2 in plasmin activation thus explains it’s importance in tumor-related angiogenesis. We verified the simultaneous upregulation of ANXA2 and PLG in S100a4-OE MH-S and cocultured HUVEC and MLE12 by immunoblotting (Figure 6D). The relevant description was added to the Results section as follows: “ANXA2 is considered to be a cellular receptor for plasminogen (PLG), often expressed on the surface of endothelial cells, macrophages, and tumor cells, which activates a cascade of pro-angiogenic factors by promoting the conversion of PLG to plasmin, thereby promoting angiogenesis and tumor progression (Semov et al., 2005; Sharma, 2019). We found synergistic upregulation of ANXA2 and PLG expression in S100a4-OE MH-S and cocultured HUVEC and MLE12, which may help explain how ANXA2 induction was involved in angiogenesis and malignant transformation (Figure 6D).” (page 14, line 338).

Recent studies showed that S100A4 is associated with tumor angiogenesis and progression by the interaction with ANXA2. ANXA2 is the endothelial receptor for S100A4 and that their interaction triggers the functional activity directly related to pathological properties of S100A4, including angiogenesis [PMID: 18608216]. It has been proved that S100A4 induces angiogenesis through interaction with ANXA2 and accelerated plasmin formation [PMID: 15788416, 25303710]. In addition, it is generally believed that ANXA2 participates in malignant cell transformation [PMID: 28867585]. Therefore, we speculate that ANXA2 may promote plasmin production by binding to S100A4, thus promoting angiogenesis and tumor initiation, and we have discussed accordingly: “The role of ANXA2 in angiogenesis has been widely recognized, and it may facilitate plasmin production by binding to S100A4 and then trigger angiogenesis and malignant cell transformation (Grindheim, Saraste, & Vedeler, 2017; Y. Liu, Myrvang, & Dekker, 2015).” (page 18, line 446).

In our study, the primary target of our validation was ANXA2 rather than RAMP1, even though its relationship with angiogenesis had been established [PMID: 20596610], so we weakened the relevant description in the manuscript.

(6) For the in vitro studies the authors might consider using primary tumor cells and not cell lines.

Thank you for your suggestion, which was in our initial experimental plan. However, since S100A4 is not expressed on the cell surface, FACS sorting of primary subset of alveolar macrophages presents technical limitations. We have also attempted overexpression in primary macrophages, but the current overexpression efficiency and cell status are not sufficient to support a subsequent series of experiments. For all these reasons, the alveolar macrophage cell line MH-S and the lung epithelial cell line MLE12 were selected to ensure the consistency and stability of the coculture system.

In addition, we are optimizing the experimental conditions to achieve coculture of primary macrophages and epithelial cells, and will also establish transgenic mouse models for simultaneous validation. The Discussion has been added as: “Besides, as our previous in vitro results were obtained based on cell lines, we will optimize the experimental conditions to achieve coculture of primary macrophage subset and epithelial cells and establish transgenic mouse models for in vivo validation.” (page 19, line 475).

**Reviewer #2 (Public review):**
Summary:The work aims to further understand the role of macrophages in lung precancer/lung cancer evolutionStrengths:(1) The use of single-cell RNA seq to provide comprehensive characterisation.(2) Characterisation of cross-talk between macrophages and the lung precancerous cells.(3) Functional validation of the effects of S100a4+ cells on lung precancerous cells using in vitro assays.(4) Validation in human tissue samples of lung precancer / invasive lesions.

We are grateful for your constructive comments. These comments are very helpful for revising and improving our paper and have provided important guiding significance to our study. We have made revisions according to your comments and have provided point-by-point responses to your concerns.

Weaknesses:(1) The authors need to provide clarification of several points in the text.

Thank you for your comment. We have clarified these points in the manuscript and responded to all your concerns in detail. Please see the responses to Recommendations for the authors.

(2) The authors need to carefully assess their assumptions regarding the role of macrophages in angiogenesis in precancerous lesions.

Thank you for your comment. We have cited relevant literature to support the occurrence of angiogenesis in precancerous lesions, and demonstrated the contribution of S100a4^+^ alveolar macrophages by tube formation assay and cytokine assay. In addition, we have discussed the relevant limitations of this study and aimed to provide more robust evidence. Please see the responses to Recommendations for the authors.

(3) The authors should discuss more broadly the current state of anti-macrophage therapies in the clinic.

Thank you for your suggestion. We have provided extensive discussion of the clinical state of anti-macrophage therapies. Please see the responses to Recommendations for the authors.

**Recommendations for the authors:**

**Reviewer #1 (Recommendations for the authors):**
The text has grammatical and syntax errors that need to be corrected accordingly.

Thank you for your suggestion. We have corrected the grammatical and syntactic errors and asked a native English speaker in the field to help polish the full text.

**Reviewer #2 (Recommendations for the authors):**
This work provides an important contribution to our further understanding of the role of macrophages in lung precancer/lung cancer evolution. I have several comments regarding how the manuscript could be improved:Introduction:The authors may consider citing the following work to enhance their work:(1) At line 78, where they talk about precancerous lesions being reversible, they should cite recent work on this in lung cancer: Teixeria et al 2019 PMID: 30664780, and Pennycuik et al 2020 PMID: 32690541.

Thank you for your suggestion. We have cited the above references in the corresponding paragraph (page 4, line 76).

(2) At line 96, where they talk about developing medicines for precancerous lesions, the authors should cite comprehensive review articles where this concept has been discussed in depth, for example: Reynolds et al 2023 PMID: 37067191, and Asad et al 2012 PMID: 23151603.

Thank you for your suggestion. We have cited the above references in the corresponding paragraph (page 5, line 94).

Results:(1) Line 142, the authors say "mice were feed for 12-16 months" - do they mean the mice were maintained for 12-16 months?

Thank you for your comment. To best mimic the process of human lung cancer development, A/J mice with the highest incidence of spontaneous lung tumors, which increases substantially with age, were selected. The corresponding description has been modified as: “A/J mice have the highest incidence of spontaneous lung tumors among various mouse strains, and this probability significantly increased with age (Landau, Wang, Yang, Ding, & Yang, 1998). To more comprehensively mirror the tumor initiation and progression process of human lung cancer, A/J mice were maintained for 12-16 months for spontaneous lesions, which resulted in three recognizable precancerous lesions in the lung.” (page 7, line 138).

(2) Line 143, the authors claim to have seen "three recognizable precancerous and cancerous lesions in the lung" but then, they only go on to describe AAH, adenoma, and AIS, lesions which are all commonly recognized as precancers. What was the cancerous (i.e. invasive) lesion they identified?

Thank you for your comment. We apologize for this misstatement and will include cancerous lesions from mice for simultaneous analysis in subsequent study. The corresponding description has been revised as: “To more comprehensively mirror the tumor initiation and progression process of human lung cancer, A/J mice were maintained for 12-16 months for spontaneous lesions, which resulted in three recognizable precancerous lesions in the lung.” (page 7, line 140).

(3) Line 172, the authors say that the "proportion of cell types across the four stages showed a dynamic trend" ... what does this mean? A trend towards what exactly?

Thank you for your comment. Our intention was to highlight heterogeneous changes, and the description has been corrected: “The proportion of cell types across the four stages showed irregular changes, while transcriptional homogeneity was reduced with precancerous progression, illustrating the importance of heterogeneity in tumorigenesis and also proving the reliability of the sampling in this study.” (page 8, line 169).

(4) Line 193, the authors say cell communication "showed a tendency to malignant transformation." What does this statement mean? If they mean more cell communication occurred in the malignant lesions than the precancerous, then there is a flaw in the logic because AAH, adenoma, and AIS are all precancerous lesions. What is the sequence of evolution to malignancy the authors are assuming? Do they mean AIS is a more advanced stage of precancerous malignancy than adenoma, and adenoma is more advanced than AAH (albeit they are all precancerous lesions).

Thank you for your comments. The malignant transformation process involves multiple stages, and histological AAH is regarded as the beginning of this process. Precancerous lesions of LUAD in mice are believed to develop stepwise from AAH, adenoma, to AIS, even if the process is not necessarily completely consistent [PMID: 11235908, 32707077]. What we meant to describe was a gradual increase in the frequency of cell communication during this process. The corresponding description has been modified as: “At the evolutionary stages of precancerous LUAD, despite possible sample heterogeneity and other interference, we observed increased interactions between epithelial cells and surrounding stromal and immune cells in the microenvironment, indicating gradually frequent cell-cell communication during this process” (page 8, line 187).

(5) Immunofluorescence images in Figure 3G and Figure 4F are captured at low magnification, making it very difficult to evaluate the colocalisation data. Suggest authors provide higher magnification images.

Thank you for your suggestion. We have replaced the immunofluorescence images in Figure 3G and Figure 4F with higher magnification images.

(6) Line 284 when referencing the cell line here, the author should make it clear in the text that cells were transfected with a construct expressing S100A4. If possible, would be good to understand if the level of S100A4 expression achieved is less, similar, or greater than that seen in these cells in vivo.

Thank you for your suggestion. We have amended the text to make it clear: “S100a4-overexpressed (OE) alveolar macrophages were established by transfection of the mS100a4 vector into the murine MH-S cell line, and empty vector was transfected as negative control (NC) cells” (page 12, line 284), and it will be clarified in the following exploration whether the level of S100a4 expression achieved is less, similar, or greater than that seen in these cells in vivo.

(7) Line 285 - when the authors first refer to OE cells that have been transfected, they should also inform the reader what NC cells are i.e. negative control cells?

Thank you for your suggestion. We have revised the relevant content as follows: “S100a4-overexpressed (OE) alveolar macrophages were established by transfection of the mS100a4 vector into the murine MH-S cell line, and empty vector was transfected as negative control (NC) cells” (page 12, line 284).

(8) Line 324 - the authors claim they have demonstrated that the macrophages promote angiogenesis through upregulation of fatty acid metabolism. Whilst they may have demonstrated changes in fatty acid metabolism, no experiments assessing the effect of the macrophages in angiogenesis assays are included in the paper, so the authors should modify this statement.

Thank you for your comments. The relevant experiments have been added based on your suggestions. Firstly, we demonstrated in vitro the up-regulation of fatty acid metabolism in S100a4^+^ alv-macro and uncovered the contribution of CPT1A to angiogenesis and cell transformation through rescue experiments; Then, HUVEC tube formation assay and cytokine assay confirmed the pro-angiogenic effect of S100a4^+^ alv-macro. We have added the Results section S100a4^+^ alv-macro drove angiogenesis by promoting Cpt1a-mediated fatty acid metabolism (page 13, line 327) and added the Discussion as: “We demonstrated the regulation of fatty acid metabolism by CPT1A in S100a4^+^ alv-macro as well as the involvement of PPAR-γ. Nevertheless, the molecular mechanism that drives the acquisition of metabolic and functional switching properties specific to this cell state still requires further characterization in the context of precancerous lesions. It has been reported that CD36 is the main effector of the S100A4/PPAR-γ pathway, and its mediated fatty acid uptake plays an important role in the tumor-promoting function of macrophages (S. Liu et al., 2021).” (page 18, line 433).

All method details covered in this section have been supplemented in the Materials and methods.

(9) Regarding angiogenesis in precancerous lesions and the role of macrophages in this process: is there even any evidence that precancerous LUAD lesions are angiogenic? Don't these lesions typically have a lepidic pattern, wherein the cancer cells merely co-opt pre-existing alveolar capillaries without the need to generate new vessels?

Thank you for your comments. As you mentioned, pathologically, precancerous LUAD lesions mainly show a lepidic growth pattern, characterized by the growth of type II alveolar epithelial cells along pre-existing alveolar walls [PMID: 29690599], but this does not mean that this process does not require the formation of new blood vessels. There are multiple patterns of tumor angiogenesis. Some studies have shown that increased angiogenesis can be observed in certain precancerous lesions, which suggests that angiogenesis may play an important role in the early stages of lung cancer development. Microvessel density (MVD) was increased in AAH and AIS compared to normal lung tissue, indicating that new blood vessels are forming to provide essential nutrients and oxygen to tumor cells to support their growth. The expression level of pro-angiogenic factors such as VEGF is usually upregulated, which promotes the formation of new blood vessels by stimulating endothelial cell proliferation and migration. [PMID: 39570802, 14568684] In addition, the infiltration of macrophages into precancerous areas in response to cytokines has been shown to trigger a tumor angiogenic switch and maintain tumor-associated continuous angiogenesis [PMID: 35022204]. Our in vitro tube formation assay and cytokine assay also demonstrated angiogenesis induced by S100a4^+^ alv-macro. We have discussed the relevant content (page 19, line 449) and will provide more sufficient evidence in future work.

Discussion:Perhaps the authors can cite any literature pertaining to the current wave of anti-macrophage therapies currently being tested in the clinic. Moreover, have these therapies been tested in lung cancer, and if so, what were the results?

Thank you for your suggestion. At present, the clinical trials of anti-macrophage therapies mainly involve Gaucher's disease and hematological malignancies, and the two tests related to lung cancer have no valid data posted. Nevertheless, there are some preclinical studies worth learning from. We have cited the relevant literature and discussed in detail: “With the elaborate resolution of TME, macrophage-related therapy is considered to be promising. So far, macrophage-targeted therapy has demonstrated clinical efficacy in Gaucher's disease and advanced hematological malignancies (Barton et al., 1991; Ossenkoppele et al., 2013). In lung cancer, an attempt to enhance anti-PD-1 therapy in NSCLC by depleting myeloid-derived suppressor cells with gemcitabine was prematurely terminated because of insufficient data collected; another clinical trial of TQB2928 monoclonal antibody promoting macrophage phagocytosis of tumor cells in combination with a third-generation EGFR TKI for advanced NSCLC is now recruiting. Moreover, preclinical studies on macrophage-targeted therapy combined with immune checkpoint inhibitors are being extensively conducted in NSCLC, and it was suggested that blockade of purine metabolism can reverse macrophage immunosuppression, and a synergetic effect can be achieved when combined with anti-PD-L1 therapy, which inspired the direction of our early intervention strategies (H. Wang, Arulraj, Anbari, & Popel, 2024; Yang et al., 2025).” (page 20, line 479).

Methods:Further description of how lesions were classified as precancerous (AAH, adenoma, AIS) or cancerous by the pathologist should be defined (or cite appropriate reference where this is described).

Thank you for your suggestion. We have cited relevant references in the Methods section (page 21, line 528) on how lesions were classified by the pathologists [PMID: 21252716, 28951454, 32707077, 24811831].